# Reinforcing General Reasoning Without Verifiers

**Xiangxin Zhou**[*,2,3], **Zichen Liu**[*,1,4], **Anya Sims**[*,1,5], **Haonan Wang**[1,4], **Tianyu Pang**[1],
**Chongxuan Li**[6], **Liang Wang**[2,3], **Min Lin**[1], **Chao Du**[‡,1]
[1]Sea AI Lab   [2]UCAS   [3]CASIA   [4]NUS   [5]Oxford   [6]RUC   ⭘ **Code Link**

## Abstract

The recent paradigm shift towards training large language models (LLMs) using DeepSeek-R1-Zero-style reinforcement learning (RL) on verifiable rewards has led to impressive advancements in code and mathematical reasoning. However, this methodology is limited to tasks where rule-based answer verification is possible and does not naturally extend to real-world domains such as chemistry, healthcare, engineering, law, biology, business, and economics. Current practical workarounds use an additional LLM as a model-based verifier; however, this introduces issues such as reliance on a strong verifier LLM, susceptibility to reward hacking, and the practical burden of maintaining the verifier model in memory during training. To address this and extend DeepSeek-R1-Zero-style training to general reasoning domains, we propose a verifier-free method (**VeriFree**) that bypasses answer verification and instead directly maximizes the probability of generating the reference answer, derived in a principled way from the RL objective. We compare VeriFree with verifier-based methods and demonstrate that, in addition to its significant practical benefits and reduced compute requirements, VeriFree matches and even surpasses verifier-based methods on extensive evaluations across MMLU-Pro, GPQA, SuperGPQA, and math-related benchmarks.

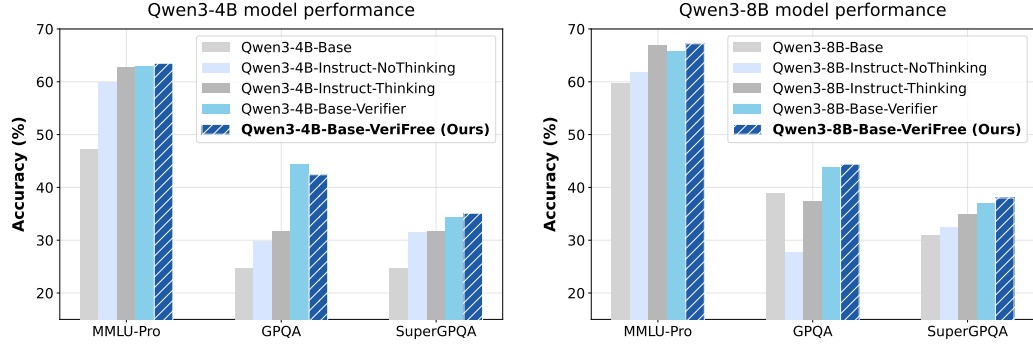

Figure 1: The general reasoning capability significantly improves when we apply VeriFree to fine-tune Qwen3 base models on a general reasoning dataset. Notably, VeriFree can match or even surpass the instruct models and the models RL-tuned from base with a specialized LLM verifier.

## 1 Introduction

DeepSeek-R1-Zero (Guo et al., 2025) recently demonstrated that training large language models (LLMs) using reinforcement learning (RL) with verifiable rewards can be extremely effective in improving reasoning capabilities. In this RL with verifiable rewards (RLVR) framework (Lambert et al., 2024), the LLM generates a reasoning trace (i.e., chain of thoughts, CoT) followed by a final answer. A rule-based program then extracts and evaluates the final answer, assigning a reward of 1 to the response if the final answer is correct and 0 otherwise. The model is then trained with RL using GRPO (Shao et al., 2024)—a simplified variant of PPO (Schulman et al., 2017).

---

*Equal contribution. ‡Correspondence to Chao Du.

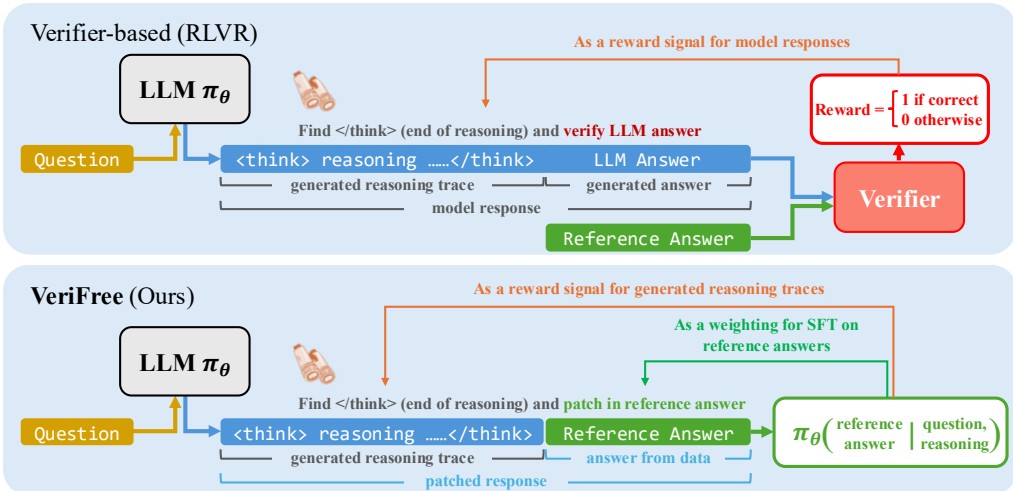

Figure 2: VeriFree enables R1-Zero-style LLM training without requiring access to a verifier. In the case of a single correct answer format, VeriFree optimizes exactly the same objective as R1-Zero with a lower variance gradient estimator.

The simplicity of this approach, coupled with impressive performance improvements in mathematical reasoning tasks, has sparked a wave of follow-up works in this paradigm of *RL with rule-based verifiable rewards* (Liu et al., 2025c; Luo et al., 2025b; Yu et al., 2025), which we will refer to as the *R1-Zero-style training* in the following. However, these methods remain limited to domains such as math and code, where rule-based verification is feasible. Reasoning is critical far beyond math and coding; however, the difficulty of answer verification in general reasoning tasks poses a major obstacle to applying this training paradigm to broader domains. To address this limitation, we investigate how to extend R1-Zero-style training to tasks where rule-based answer verification is not possible.

A natural extension, as explored in recent general reasoning works (Su et al., 2025; Ma et al., 2025), is to introduce a specialized LLM as a verifier, similar to the reward model used in RL from human feedback (RLHF) (Ziegler et al., 2019; Ouyang et al., 2022). In these methods, the *model-based verifier* is queried to determine whether the generated answer is equivalent to the reference answer. Although this approach bypasses the need for rule-based evaluation, it introduces several potential drawbacks (as in standard RLHF): it depends on the availability of a strong verifier LLM, it converts the R1-Zero-style paradigm into optimizing a model-based reward, which makes it vulnerable to reward hacking (Gao et al., 2023), and it adds computational overhead by requiring an additional model to be held in memory and queried during training.

In this work, we propose an alternative: a verifier-free approach that preserves the benefits of the RL paradigm while removing the reliance on explicit rule-based or model-based verification. Starting from a principled objective, we derive a method that directly maximizes the likelihood of generating reference answers. The gradient naturally decomposes into two terms: one analogous to RLVR with likelihood as a reward signal, and the other resembling to supervised training on reference answers. Fig. 2 illustrates our method, which we term VeriFree, as it does not rely on verifiers.

This approach has several appealing properties. First, when there is a unique correct answer string, our method is equivalent in expectation to the objective in RLVR, but with lower variance, which can be viewed as a form of reward shaping (Ng et al., 1999; Randlov & Alstrøm, 1998). Even when multiple valid answers exist, we show empirically that using just one as a reference provides a sufficient learning signal to elicit strong reasoning behavior. Additionally, this framework can be viewed through a variational lens as a neat way of optimizing over latent reasoning traces.

To make this work in practice, we identify and address several subtle challenges, including effective variance reduction and precise handling of tokenization at the reasoning-answer patching point. We conduct comprehensive ablations to understand the impact of each design choice. We benchmark our method across a diverse set of general reasoning tasks, and the results are striking: as shown in Fig. 1, VeriFree not only matches but often outperforms verifier-based alternatives, while being simpler, faster, less memory-intensive, and more robust.

## 2 METHODOLOGY

### 2.1 PRELIMINARIES: VERIFIER-BASED REINFORCEMENT LEARNING

In RL applied to LLMs, the language model is treated as a policy $\pi_\theta$ that generates an output $\boldsymbol{o}$ autoregressively in response to an input question $\boldsymbol{x}$. The goal is typically to optimize $\pi_\theta$ to maximize a given reward function $R(\boldsymbol{x}, \boldsymbol{o})$:

$$\theta \in \arg\max_\theta \mathbb{E}_{\boldsymbol{o} \sim \pi_\theta(\cdot|\boldsymbol{x})} \left[ R(\boldsymbol{x}, \boldsymbol{o}) \right]. \tag{1}$$

In R1-Zero-style RL, the reward is computed by first parsing the response $\boldsymbol{o}$ into a reasoning trace $\boldsymbol{z}$ and a final answer $\boldsymbol{y}$. A verifier then checks $\boldsymbol{y}$ against the ground-truth reference answer $\boldsymbol{y}^\star$ and assigns a binary reward based on correctness, namely $R_{\text{Verifier}}(\boldsymbol{y}; \boldsymbol{y}^\star) = \mathbb{1}_{\{\boldsymbol{y} \equiv \boldsymbol{y}^\star\}}$.[1] Decomposing the model output as $\boldsymbol{o} = (\boldsymbol{z}, \boldsymbol{y})$, we can rewrite the objective in Eq. (1) as:

$$J_{\text{Verifier}}(\theta; \boldsymbol{x}, \boldsymbol{y}^\star) = \mathbb{E}_{\boldsymbol{z} \sim \pi_\theta(\cdot|\boldsymbol{x})} \mathbb{E}_{\boldsymbol{y} \sim \pi_\theta(\cdot|\boldsymbol{x}, \boldsymbol{z})} [R_{\text{Verifier}}(\boldsymbol{y}; \boldsymbol{y}^\star)], \tag{2}$$

which separates the sampling of the reasoning trace and the final answer. To maximize this objective, the model $\pi_\theta$ is typically updated using the policy gradient estimator (Sutton & Barto, 2018):

$$\nabla_\theta J_{\text{Verifier}}(\theta; \boldsymbol{x}, \boldsymbol{y}^\star) = \mathbb{E}_{\boldsymbol{z} \sim \pi_\theta(\cdot|\boldsymbol{x})} \mathbb{E}_{\boldsymbol{y} \sim \pi_\theta(\cdot|\boldsymbol{x}, \boldsymbol{z})} \Big[ R_{\text{Verifier}}(\boldsymbol{y}; \boldsymbol{y}^\star) \big[ \nabla_\theta \log \pi_\theta(\boldsymbol{z}|\boldsymbol{x}) + \nabla_\theta \log \pi_\theta(\boldsymbol{y}|\boldsymbol{x}, \boldsymbol{z}) \big] \Big]. \tag{3}$$

However, this approach requires evaluating answer correctness via $R_{\text{Verifier}}(\boldsymbol{y}; \boldsymbol{y}^\star) = \mathbb{1}_{\{\boldsymbol{y} \equiv \boldsymbol{y}^\star\}}$, which is often nontrivial. While in domains such as math and code, this evaluation is feasible via rules (Guo et al., 2025; Liu et al., 2025c) or test cases (Gehring et al., 2024), accurate verification in general reasoning tasks is substantially more difficult. As a result, recent advances in R1-Zero-style training have largely been restricted to verifiable domains, leaving reasoning tasks in general domains under-explored. In light of this, we present an alternative verifier-free approach which naturally extends this training paradigm to broader reasoning domains.

### 2.2 VERIFREE POLICY OPTIMIZATION

We begin with the standard objective in Eq. (2) and show that, in the case of a single correct answer, we can derive an equivalent objective that does not require a verifier. Moreover, we compare the gradient estimators of this new objective and the verifier-based objective (Eq. (3)), and demonstrate that our verifier-free gradient estimator has the additional benefit of lower variance.

Starting from Eq. (2) and assuming a unique correct answer such that $R_{\text{Verifier}}(\boldsymbol{y}; \boldsymbol{y}^*) = \mathbb{1}_{\{\boldsymbol{y} = \boldsymbol{y}^\star\}}$ (i.e., exact match rather than semantic equivalence $\mathbb{1}_{\{\boldsymbol{y} \equiv \boldsymbol{y}^\star\}}$), the VeriFree objective is derived as:

$$
\begin{aligned}
J_{\text{Verifier}}(\theta; \boldsymbol{x}, \boldsymbol{y}^\star) &= \mathbb{E}_{\boldsymbol{z} \sim \pi_\theta(\cdot|\boldsymbol{x})} \Big[ \mathbb{E}_{\boldsymbol{y} \sim \pi_\theta(\cdot|\boldsymbol{x}, \boldsymbol{z})} \big[ \overbrace{\mathbb{1}_{\{\boldsymbol{y} = \boldsymbol{y}^\star\}}}^{R_{\text{Verifier}}} \big] \Big] \\
&= \mathbb{E}_{\boldsymbol{z} \sim \pi_\theta(\cdot|\boldsymbol{x})} \Big[ \sum_{\boldsymbol{y}} \pi_\theta(\boldsymbol{y}|\boldsymbol{x}, \boldsymbol{z}) \mathbb{1}_{\{\boldsymbol{y} = \boldsymbol{y}^\star\}} \Big] \\
&= \mathbb{E}_{\boldsymbol{z} \sim \pi_\theta(\cdot|\boldsymbol{x})} \Big[ \underbrace{\pi_\theta(\boldsymbol{y}^\star|\boldsymbol{x}, \boldsymbol{z})}_{\triangleq R_{\text{VeriFree}}(\boldsymbol{z}; \boldsymbol{x}, \boldsymbol{y}^\star)} \Big] \triangleq J_{\text{VeriFree}}(\theta; \boldsymbol{x}, \boldsymbol{y}^\star).
\end{aligned} \tag{4}
$$

This can be interpreted as follows: if only one answer $\boldsymbol{y}^\star$ is correct and receives a reward of 1 (while all others receive 0), then the expected reward given a reasoning trace $\boldsymbol{z}$ can be computed directly as the probability assigned to $\boldsymbol{y}^\star$, effectively marginalizing out $\boldsymbol{y}$. The corresponding gradient estimator is given by (see Appendix B.1 for a full derivation):

$$\nabla_\theta J_{\text{VeriFree}}(\theta; \boldsymbol{x}, \boldsymbol{y}^\star) = \mathbb{E}_{\boldsymbol{z} \sim \pi_\theta(\cdot|\boldsymbol{x})} \Big[ R_{\text{VeriFree}}(\boldsymbol{z}; \boldsymbol{x}, \boldsymbol{y}^\star) \big[ \underbrace{\nabla_\theta \log \pi_\theta(\boldsymbol{z}|\boldsymbol{x})}_{\text{reasoning term}} + \underbrace{\nabla_\theta \log \pi_\theta(\boldsymbol{y}^\star|\boldsymbol{x}, \boldsymbol{z})}_{\text{reference answer term}} \big] \Big]. \tag{5}$$

Both the objective and its gradient estimator (Eq. (4) and (5)) are equivalent in expectation to their verifier-based counterparts (Eq. (2) and (3)). Intuitively, the "reasoning term" in Eq. (5) can be

---

[1] We use '$\equiv$' to denote semantic equivalence, where multiple expressions can be judged correct. For example, if $\boldsymbol{y}^\star = $ '8/5', then '8/5', '1.6', and '\frac{8}{5}' are all considered correct.

| **Verifier-based (R1-Zero)** | **VeriFree (Ours)** |
|---|---|
| Model generates the reasoning trace $z$ and answer $y$. | Model generates the reasoning trace $z$. |
| Extract the answer $y$. | Patch in the correct answer $y^\star$. |
| **Check answer using a verifier**. | **Evaluate probability $\pi_\theta(y^\star\|x, z)$.** |
| Reward $R_{\text{Verifier}} = 1$ if correct, 0 otherwise. | Reward $R_{\text{VeriFree}} = \pi_\theta(y^\star\|x, z)$. |
| Train with gradient estimator $\nabla_\theta J_{\text{Verifier}}$ (Eq. 3). | Train with gradient estimator $\nabla_\theta J_{\text{VeriFree}}$ (Eq. 5). |

Figure 3: A pseudocode-like comparison of VeriFree (ours) and the standard R1-Zero approach.

interpreted as a policy gradient where the reward for a reasoning trace $z$ (i.e., $R_{\text{VeriFree}}(z; x, y^\star)$) is the probability that the policy will generate the correct answer $y^\star$ given $z$, while the "reference answer term" can be viewed as a reward-weighted supervised learning term for $y^\star$ given $z$. We will further elaborate on this interpretation in Sec. 2.3. In addition to bypassing the need for a verifier, our VeriFree gradient estimator also benefits from reduced variance:

**Theorem 1.** (*Variance Reduction*) *Let $\hat{G}_{\text{Verifier}}(x, y^\star, z, y)$ and $\hat{G}_{\text{VeriFree}}(x, y^\star, z)$ denote the single-sample Monte Carlo estimators of $\nabla_\theta J_{\text{Verifier}}$ and $\nabla_\theta J_{\text{VeriFree}}$ given $x$ and $y^\star$, respectively. Then we have*

$$Var_{z \sim \pi_\theta(\cdot|x)}\left[\hat{G}_{\text{VeriFree}}(x, y^\star, z)\right] \leq Var_{z \sim \pi_\theta(\cdot|x), y \sim \pi_\theta(\cdot|x, z)}\left[\hat{G}_{\text{Verifier}}(x, y^\star, z, y)\right]. \quad (6)$$

This reduction in variance arises from Rao-Blackwellization (Casella & Robert, 1996). For intuition, the variance in the Monte Carlo estimate of $\nabla_\theta J_{\text{Verifier}}$ stems from the randomness in sampling $z \sim \pi_\theta(\cdot|x)$ and $y \sim \pi_\theta(\cdot|x, z)$, while for estimating $\nabla_\theta J_{\text{VeriFree}}$ we analytically marginalize out $y$, thereby removing this source of randomness. We provide a full proof in Appendix B.2.

Our gradient estimator $\nabla_\theta J_{\text{VeriFree}}(\theta; x, y^\star)$ is fully compatible with other variance reduction techniques, including RLOO (Ahmadian et al., 2024), GRPO (Shao et al., 2024) reward normalizations, and the PPO (Schulman et al., 2017) clipping operation. As such, we sample multiple responses for each prompt and apply the RLOO baseline to the reasoning term in Eq. (5). We also adopt the corrected response-length normalization from Liu et al. (2025c). The final on-policy gradient estimator is as follows:

$$\nabla_\theta J_{\text{VeriFree}}(\theta) = \frac{1}{G} \sum_{i=1}^{G} \left[A_i \cdot \nabla_\theta \log \pi_\theta(z_i|x) + R_i \cdot \nabla_\theta \log \pi_\theta(y^\star|x, z_i)\right], \quad (7)$$

where $z_i \sim \pi_\theta(\cdot|x)$, $R_i = \pi_\theta(y^\star|x, z_i)$, and $A_i = \pi_\theta(y^\star|x, z_i) - \frac{1}{G-1}\sum_{j \neq i} \pi_\theta(y^\star|x, z_j)$. We also provide the PPO-based off-policy variant in Appendix C.

## 2.3 COMPARISON TO EXISTING APPROACHES

There have been two main prior works that, although derived from a different perspective, arrive at related alternative gradient estimators: JEPO (Tang et al., 2025) and LaTRO (Chen et al., 2024b).

$$\nabla_\theta J_{\text{Verifier}} = \mathbb{E}_{z,y}\Big[ \mathbb{1}_{\{y \equiv y^\star\}} \overbrace{\nabla_\theta \log \pi_\theta(z|x)}^{\text{reasoning term}} + \mathbb{1}_{\{y \equiv y^\star\}} \overbrace{\nabla_\theta \log \pi_\theta(y|x, z)}^{\text{answer term}} \Big] \quad \text{(R1-Zero)}$$

$$\nabla_\theta J_{\text{VeriFree}} = \mathbb{E}_z\Big[ \pi_\theta(y^\star|x, z) \nabla_\theta \log \pi_\theta(z|x) + \pi_\theta(y^\star|x, z) \overbrace{\nabla_\theta \log \pi_\theta(y^\star|x, z)}^{\text{reference answer term}} \Big] \quad \text{(Ours)}$$

$$\nabla_\theta J_{\text{JEPO}} = \mathbb{E}_z\Big[ \log \pi_\theta(y^\star|x, z) \nabla_\theta \log \pi_\theta(z|x) + 1 \cdot \nabla_\theta \log \pi_\theta(y^\star|x, z) \Big] \quad \text{(JEPO)}$$

$$\nabla_\theta J_{\text{LaTRO}} = \mathbb{E}_z\Big[ \big(\log \pi_\theta(y^\star|x, z) - \log \tfrac{\pi_\theta(z|x)}{\pi_{\text{ref}}(z|x)}\big) \nabla_\theta \log \pi_\theta(z|x) + 1 \cdot \nabla_\theta \log \pi_\theta(y^\star|x, z) \Big] \quad \text{(LaTRO)}$$

Both JEPO and LaTRO treat the reasoning trace $z$ as a latent variable and extend the standard supervised learning objective (log-likelihood) to optimize lower bounds on $\log(\mathbb{E}_{z \sim \pi_\theta(\cdot|x)}\left[\pi_\theta(y^\star|x, z)\right])$ and $\log(\mathbb{E}_{z \sim \pi_{\text{ref}}(\cdot|x)}\left[\pi_\theta(y^\star|x, z)\right])$, respectively. The primary difference is that JEPO samples $z$ from the learned policy $\pi_\theta$, while LaTRO uses a fixed reference policy $\pi_{\text{ref}}$. Despite originating

from different perspectives, these methods arrive at similar gradient estimators, as shown in above comparisons. However, as reported in Tang et al. (2025), these verifier-free, variational-inference-based methods consistently underperform the standard verifier-based R1-Zero approach. In contrast, our method matches or outperforms the verifier-based baselines. We also include experimental comparison with JEPO and LaTRO in Appendix E.2.

One possible explanation is that our method *exactly* recovers the original verifier-based objective under the single-correct-answer assumption, whereas JEPO and LaTRO optimize subtly different objectives. For example, JEPO effectively uses a reward of $R = \log \pi_\theta(\boldsymbol{y}^\star|\boldsymbol{x}, \boldsymbol{z})$, as highlighted in the gradient expressions above. Another distinction lies in the weighting of the "reference answer term" $\nabla_\theta \log \pi_\theta(\boldsymbol{y}^\star|\boldsymbol{x}, \boldsymbol{z})$. In our method, this term is weighted by the probability $\pi_\theta(\boldsymbol{y}^\star|\boldsymbol{x}, \boldsymbol{z})$, which is the likelihood of the reference answer given the sampled reasoning trace. In contrast, both JEPO and LaTRO use a fixed weight of 1, thereby increasing the probability of $\boldsymbol{y}^\star$ regardless of the quality of the reasoning trace $\boldsymbol{z}$. We hypothesize that this behavior could promote poor reasoning. For instance, suppose the model generates the reasoning trace "`... minus 2 apples, finally resulting in a total of 7 apples`" when the correct answer is "6". The JEPO and LaTRO objectives would still push the model to output "6" from that flawed trace, reinforcing a mismatch between reasoning and answer. Our method avoids this by down-weighting contributions from low-quality traces. Due to space constraints, more related work is deferred to Appendix A.

### 2.4 How to handle the tokenization at patching point?

A critical consideration when extracting reasoning traces $\boldsymbol{z}$ from model responses $(\boldsymbol{z}, \boldsymbol{y})$ stems from the fact that LLMs operate on token sequences, not raw text strings. While human-readable outputs (e.g., "`...<answer> \\boxed{...} </answer>`" as in Template 1) suggest splitting reasoning traces $\boldsymbol{z}$ at specific text patterns like "`<answer>`", such text-based splitting strategy may cause tokenization inconsistencies. For example, the "`>`" character might be tokenized differently depending on its surrounding context in $\boldsymbol{y}$ versus $\boldsymbol{y}^\star$. While one potential solution is to introduce special tokens to enforce consistent tokenization boundaries, these novel tokens could harm model performance due to their absence from the base model's pretraining vocabulary.

Instead, we resolve this by defining the end of $\boldsymbol{z}$ at the token corresponding to `<answer` (i.e., without "`>`"), leveraging the fact that the pattern "`r>`" does not appear in standard tokenizer vocabularies. This ensures consistent token-space alignment between sampling and optimization, avoiding instability due to off-policy mismatches (Yao et al., 2025). Notably, this approach is operationally equivalent to setting "`<answer`" (instead of "`<answer>`") as the stop word during sampling, a mechanism natively supported by modern inference engines like vLLM (Kwon et al., 2023). In this case, we can sample reasoning traces $\boldsymbol{z}$ directly, instead of first generating the full response $(\boldsymbol{z}, \boldsymbol{y})$ and then extracting $\boldsymbol{z}$ post hoc.

## 3 Experiments

In this section, we empirically evaluate VeriFree across multiple model scales and reasoning benchmarks. We show that VeriFree improves general reasoning capabilities over verifier-based baselines, achieves higher learning efficiency, transfers to math reasoning tasks without math-specific supervision, and benefits from its key design choices, including tokenization-aware reasoning trace extraction and the RLOO variance reduction technique.

### 3.1 Setup

**Training.** Following the "Zero" setting widely adopted in recent work (Guo et al., 2025; Hu et al., 2025; Liu et al., 2025c; Ma et al., 2025), we directly fine-tune the base LLM, skipping the intermediate stage of supervised fine-tuning (SFT). We implement our training pipeline using Oat (Liu et al., 2024) by instantiating their base modules and incorporate our algorithmic changes. Our experiments are conducted using Qwen3 (Team, 2025) base models across multiple scales, including 1.7B, 4B, and 8B parameters. We adopt the prompt template shown in Template 1. We do not employ KL regularization losses or KL penalties in rewards, as recent studies suggest that removing KL terms does not have a significant impact (Liu et al., 2025c; Hu et al., 2025). As a result, our method does not require maintaining a reference model in memory.

Table 1: Accuracy comparison on the MMLU-Pro benchmark.

| Method | Len. | Avg. | CS | Math | Chem | Eng | Law | Bio | Health | Phys | Bus | Phil | Econ | Other | Psy | Hist |
|---|---|---|---|---|---|---|---|---|---|---|---|---|---|---|---|---|
| Qwen3-1.7B-Base | 618 | 33.3 | 34.6 | 38.9 | 32.2 | 21.0 | 17.3 | 56.1 | 33.5 | 32.0 | 38.5 | 21.8 | 45.7 | 28.4 | 44.4 | 21.0 |
| Qwen3-1.7B (w/o thinking) | 893 | 46.1 | 49.5 | 64.4 | 48.0 | 35.9 | 22.9 | 64.9 | 38.0 | 49.7 | 53.5 | 33.7 | 53.9 | 36.4 | 51.6 | 31.2 |
| Qwen3-1.7B (w/ thinking) | 3904 | 52.0 | 56.1 | 76.4 | 57.6 | 27.0 | 21.9 | 67.9 | 47.7 | 57.5 | 61.3 | 38.9 | 64.5 | 42.5 | 59.2 | 32.3 |
| Qwen3-1.7B-Base-Verifier | 875 | **47.0** | 48.8 | 64.4 | 52.7 | 38.1 | 18.7 | 62.9 | 41.2 | 51.9 | 54.9 | 31.9 | 55.2 | 38.6 | 53.3 | 30.2 |
| Qwen3-1.7B-Base-VeriFree | 856 | 46.9 | 46.8 | 64.1 | 51.7 | 41.8 | 20.0 | 64.0 | 39.7 | 52.1 | 55.6 | 29.5 | 53.1 | 37.5 | 53.0 | 29.9 |
| Qwen3-4B-Base | 825 | 47.2 | 42.9 | 67.1 | 55.5 | 40.0 | 22.5 | 56.9 | 43.6 | 55.4 | 54.9 | 27.5 | 52.7 | 34.3 | 48.6 | 34.7 |
| Qwen3-4B (w/o thinking) | 838 | 60.0 | 65.9 | 79.1 | 65.8 | 45.7 | 29.0 | 76.6 | 57.0 | 65.1 | 66.7 | 48.9 | 69.2 | 52.1 | 64.3 | 44.6 |
| Qwen3-4B (w/ thinking) | 3456 | 62.7 | 70.0 | 84.8 | 66.6 | 38.6 | 28.7 | 81.3 | 60.4 | 67.4 | 69.2 | 53.7 | 75.1 | 57.8 | 67.9 | 49.6 |
| Qwen3-4B-Base-Verifier | 921 | 63.0 | 66.1 | 81.3 | 69.7 | 52.8 | 29.1 | 79.8 | 62.8 | 67.6 | 71.2 | 48.5 | 73.1 | 52.8 | 68.5 | 45.4 |
| Qwen3-4B-Base-VeriFree | 1241 | **63.5** | 64.4 | 82.2 | 70.1 | 55.6 | 30.7 | 81.7 | 59.2 | 71.0 | 71.0 | 47.1 | 71.7 | 53.4 | 66.8 | 47.5 |
| Qwen2.5-7B | 519 | 47.8 | 48.3 | 59.5 | 44.4 | 33.4 | 25.1 | 63.6 | 50.4 | 48.0 | 55.9 | 34.7 | 60.6 | 46.0 | 58.2 | 38.3 |
| Qwen2.5-7B-SimpleRL-Zoo | 705 | 51.2 | 51.2 | 52.0 | 50.2 | 40.8 | 30.5 | 69.5 | 54.3 | 52.5 | 57.3 | 41.9 | 62.8 | 52.6 | 60.8 | 42.3 |
| Qwen2.5-Math-7B-Oat-Zero | 556 | 40.5 | 47.6 | 47.7 | 46.9 | 32.1 | 18.1 | 53.6 | 25.7 | 49.4 | 52.9 | 29.5 | 54.7 | 32.8 | 43.0 | 22.8 |
| Qwen2.5-7B-Instruct | 481 | 55.3 | 56.6 | 70.4 | 55.6 | 42.7 | 29.8 | 69.3 | 55.1 | 57.9 | 63.5 | 41.5 | 63.4 | 53.6 | 62.4 | 43.6 |
| General-Reasoner-7B | 867 | 58.7 | 63.4 | 73.7 | 63.3 | 44.9 | 35.2 | 72.0 | 56.6 | 61.5 | 66.7 | 43.1 | 68.1 | 52.8 | 62.8 | 47.8 |
| Qwen3-8B-Base | 613 | 59.8 | 61.2 | 75.0 | 66.2 | 46.7 | 31.4 | 75.9 | 60.4 | 62.1 | 65.9 | 48.7 | 69.0 | 54.3 | 63.9 | 47.2 |
| Qwen3-8B (w/o thinking) | 1032 | 61.9 | 65.6 | 71.9 | 62.8 | 46.2 | 34.7 | 79.9 | 66.1 | 63.7 | 69.3 | 55.9 | 72.9 | 58.9 | 67.9 | 52.5 |
| Qwen3-8B (w/ thinking) | 3952 | 66.9 | 71.5 | 83.8 | 68.0 | 38.7 | 39.2 | 85.2 | 72.1 | 69.8 | 73.3 | 57.5 | 79.2 | 66.3 | 71.8 | 57.7 |
| Qwen3-8B-Base-Verifier | 594 | 65.9 | 63.9 | 81.8 | 71.1 | 56.9 | 35.4 | 81.9 | 64.9 | 71.6 | 74.1 | 53.9 | 74.2 | 58.4 | 68.4 | 54.3 |
| Qwen3-8B-Base-VeriFree | 776 | **67.2** | 71.5 | 85.3 | 73.5 | 55.7 | 37.3 | 81.9 | 64.3 | 73.1 | 74.1 | 54.9 | 74.8 | 59.6 | 67.7 | 54.6 |

---

**Template 1 (for Ours).**

```
<|im_start|>user\n{question}\nPlease reason step by step,
and put your final answer within <answer> \\boxed{}
</answer>.<|im_end|>\n<|im_start|>assistant\n
```

---

For the 1.7B and 4B models, we conduct fine-tuning for approximately 4,000 policy gradient steps; for the 8B models, we fine-tune for around 3,000 policy gradient steps. During each step, the policy model (i.e., the LLM) generates 8 responses for each question (i.e., `group_size=8`), with 16 questions processed per step. We use the sampling configurations `temperature=1.0`, `top_p=1`, and `max_tokens=3000` for the rollout process. The responses are then parsed into reasoning traces and model-predicted answers. We replace the model-predicted answers with the reference answers from the training dataset. Subsequently, a single forward pass is executed to compute the conditional probability of the reference answer, conditioned on all preceding tokens including the prompt and the reasoning trace. This procedure introduces only a minimal additional computational cost, as the forward pass of the LLM does not require autoregressive decoding and does not require storing intermediate states for backpropagation. All collected samples from each step are used for one optimization step. The training is conducted on a single node with 8×H100 GPUs.

**Dataset.** To support general reasoning, we begin with the dataset curated by Ma et al. (2025), sourced from WebInstruct (Yue et al., 2024). To improve data quality and reliability and reduce size, we retain samples with answers that consist of fewer than seven tokens, and use Qwen2.5-72B-Instruct (Yang et al., 2024a) to filter out low-quality and noisy data. This process results in approximately 61,000 data samples spanning diverse domains, which we refer to as **WebData**. The category distribution is visualized in Fig. 7.

**Evaluation.** In line with prior work (Ma et al., 2025), we employ multiple-choice questions for evaluation to facilitate verification. To assess general reasoning abilities, we utilize the following benchmarks: **MMLU-Pro** (Wang et al., 2024), a challenging multi-task understanding benchmark designed to evaluate the capabilities of LLMs across various domains; **SuperGPQA** (Du et al., 2025), a large-scale benchmark consisting of graduate-level reasoning questions spanning 285 diverse disciplines; and **GPQA-Diamond** (Rein et al., 2024), which focuses on graduate-level question-answering and is designed to resist shallow pattern-matching and memorization. While our primary focus is not on enhancing mathematical abilities, we also evaluate math reasoning using a suite of standard math reasoning benchmarks. This suite includes **MATH-500** (Hendrycks et al., 2021), **OlympiadBench** (He et al., 2024), **Minerva Math** (Lewkowycz et al., 2022), **GSM8K** (Cobbe

Table 2: Accuracy comparison on the SuperGPQA benchmark.

| Method | Len. | Avg. | Eng. | Med. | Sci. | Phil. | M.S. | Econ. | Mgmt. | Socio. | L/A | Hist. | Agron. | Law | Edu. |
|---|---|---|---|---|---|---|---|---|---|---|---|---|---|---|---|
| Qwen3-1.7B-Base | 997 | 17.4 | 17.7 | 18.6 | 16.0 | 27.4 | 27.3 | 20.5 | 22.4 | 23.1 | 15.5 | 11.1 | 18.1 | 20.6 | 21.3 |
| Qwen3-1.7B (w/o thinking) | 1152 | 23.3 | 22.6 | 22.8 | 24.3 | 30.3 | 31.2 | 24.3 | 27.5 | 23.8 | 19.0 | 18.1 | 20.8 | 24.5 | 28.1 |
| Qwen3-1.7B (w/ thinking) | 4799 | 23.6 | 21.8 | 25.3 | 23.6 | 33.1 | 33.7 | 29.6 | 27.5 | 32.2 | 19.0 | 18.0 | 25.0 | 26.2 | 31.6 |
| Qwen3-1.7B-Base-Verifier | 1049 | 24.5 | 26.0 | 23.9 | 24.4 | 30.8 | 26.8 | 26.9 | 27.0 | 26.6 | 18.9 | 16.6 | 22.3 | 22.6 | 27.3 |
| Qwen3-1.7B-Base-VeriFree | 964 | 24.8 | 25.7 | 24.7 | 24.9 | 26.5 | 30.2 | 25.9 | 27.9 | 28.0 | 20.4 | 15.9 | 22.9 | 25.0 | 28.9 |
| Qwen3-4B-Base | 902 | 24.7 | 25.7 | 23.6 | 26.0 | 23.6 | 25.4 | 28.8 | 28.4 | 19.6 | 16.4 | 16.8 | 20.6 | 25.6 | 24.0 |
| Qwen3-4B (w/o thinking) | 1397 | 31.6 | 32.0 | 31.5 | 32.3 | 37.5 | 36.1 | 37.8 | 33.7 | 33.6 | 24.3 | 20.6 | 28.3 | 31.4 | 33.5 |
| Qwen3-4B (w/ thinking) | 4568 | 31.7 | 30.7 | 33.2 | 32.1 | 41.2 | 31.7 | 41.7 | 35.9 | 32.9 | 24.5 | 22.4 | 30.9 | 35.7 | 35.3 |
| Qwen3-4B-Base-Verifier | 1045 | 34.3 | 35.4 | 35.5 | 34.5 | 39.2 | 41.0 | 39.1 | 36.7 | 37.1 | 26.6 | 22.3 | 33.8 | 33.1 | 35.3 |
| Qwen3-4B-Base-VeriFree | 1451 | 35.1 | 36.3 | 34.5 | 36.9 | 35.7 | 37.1 | 39.1 | 38.3 | 31.5 | 24.7 | 22.0 | 33.0 | 33.2 | 34.1 |
| Qwen2.5-7B | 716 | 23.8 | 24.2 | 27.0 | 21.8 | 28.8 | 31.2 | 27.6 | 29.1 | 22.4 | 20.8 | 20.2 | 24.5 | 27.4 | 30.2 |
| Qwen-2.5-7B-SimpleRL-Zoo | 850 | 26.3 | 26.4 | 30.5 | 23.8 | 32.6 | 32.2 | 33.0 | 31.9 | 28.7 | 24.1 | 21.4 | 27.2 | 29.6 | 32.9 |
| Qwen2.5-Math-7B-Oat-Zero | 638 | 21.3 | 23.1 | 16.4 | 21.5 | 23.1 | 21.5 | 25.9 | 27.2 | 25.2 | 17.7 | 15.9 | 21.4 | 18.8 | 24.8 |
| Qwen2.5-7B-Instruct | 604 | 28.4 | 27.7 | 32.2 | 27.6 | 33.7 | 32.2 | 32.4 | 32.9 | 32.9 | 24.5 | 22.1 | 29.7 | 30.6 | 32.4 |
| General-Reasoner-7B | 1047 | 30.8 | 31.5 | 32.2 | 29.9 | 35.2 | 41.5 | 38.4 | 33.1 | 35.0 | 25.5 | 22.7 | 28.9 | 32.5 | 35.5 |
| Qwen3-8B-Base | 825 | 31.0 | 31.3 | 34.0 | 30.6 | 36.0 | 37.1 | 34.7 | 37.5 | 35.0 | 24.2 | 20.0 | 28.5 | 31.4 | 36.4 |
| Qwen3-8B (w/o thinking) | 1638 | 32.4 | 32.6 | 36.5 | 31.2 | 39.5 | 42.0 | 37.7 | 37.3 | 38.5 | 25.0 | 22.6 | 33.2 | 34.3 | 38.4 |
| Qwen3-8B (w/ thinking) | 4995 | 35.0 | 33.6 | 42.1 | 33.5 | 44.4 | 37.6 | 44.2 | 42.7 | 42.7 | 27.9 | 24.9 | 37.9 | 38.7 | 40.7 |
| Qwen3-8B-Base-Verifier | 713 | 37.1 | 38.2 | 39.5 | 37.2 | 39.5 | 39.5 | 43.0 | 40.1 | 38.5 | 28.9 | 24.8 | 34.2 | 34.8 | 38.2 |
| Qwen3-8B-Base-VeriFree | 951 | 38.0 | 38.3 | 39.1 | 39.6 | 37.5 | 42.9 | 41.8 | 41.7 | 44.8 | 28.6 | 23.3 | 33.6 | 36.3 | 38.6 |

et al., 2021), **AMC** and **AIME24** (Li et al., 2024). We utilize Math-Verify[2] to check for answer equivalence. Except for AIME24, where we employ a `temperature=1.0` and repeat each question 32 times, all other benchmarks are evaluated using `temperature=0.0`. We use `max_tokens=8192` for all evaluations. For the multiple-domain benchmarks (MMLU-Pro and SuperGPQA), we report a micro-average. This metric computes a global average by aggregating contributions across all classes, rather than calculating a macro-average (which would average per-class scores and assign equal weight to each domain).

**Baselines.** Our primary baseline, denoted **Verifier**, is a verifier-based approach using the verifier from Ma et al. (2025). The verifier is initialized from Qwen2.5-Math-1.5B (Yang et al., 2024b) and fine-tuned on data generated by Gemini 2.0 Flash to assess equivalence between the reference and predicted answers, conditioned on the question. We apply Dr. GRPO (Liu et al., 2025c) as the optimization algorithm for the baseline, ensuring that all other settings are consistent with our approach. Following Ma et al. (2025), the reward definition incorporates additional factors beyond verifier correctness, including format compliance and the length of generated answers. If the format is incorrect (e.g., missing `\\boxed{}` in the model response), a negative reward of -0.5 is applied. Moreover, a length penalty of `-0.05 × min(10, abs(length_of_correct_answer - length_of_answer))` is added.

We also report the results for the base and the instruct models of Qwen-3-1.7B/4B/8B (Team, 2025) and Qwen2.5-7B (Yang et al., 2024a), as well as the checkpoints released by Qwen2.5-7B-SimpleRL-Zoo (Zeng et al., 2025), Qwen2.5-Math-7B-Oat-Zero (Liu et al., 2025c), and General-Reasoner-7B (Ma et al., 2025).[3] Notably, Qwen3 integrates both a *thinking* mode (for complex, multi-step reasoning) and a *non-thinking* mode (for rapid, context-driven responses) within a unified framework. We report results of both modes on Qwen3 instruct models.

## 3.2 MAIN RESULTS

**VeriFree improves general reasoning capabilities.** We begin by evaluating the effectiveness of VeriFree in enhancing the general reasoning capabilities of LLMs using the MMLU-Pro and SuperGPQA benchmarks. Table 1 presents a detailed comparison across model scales and domains in the MMLU-Pro benchmark. Starting from base models, applying RL with VeriFree yields substan-

---

[2] https://github.com/huggingface/Math-Verify

[3] Qwen2.5 and Qwen3 use different naming conventions in their official release. For base models: Qwen2.5 has no suffix (e.g., Qwen2.5-7B), whereas Qwen3 adds "-Base" (e.g., Qwen3-8B-Base). For instruct models: Qwen2.5 uses "-Instruct" (e.g., Qwen2.5-7B-Instruct), while Qwen3 omits the suffix (e.g., Qwen3-8B). We follow these conventions consistently in the paper.

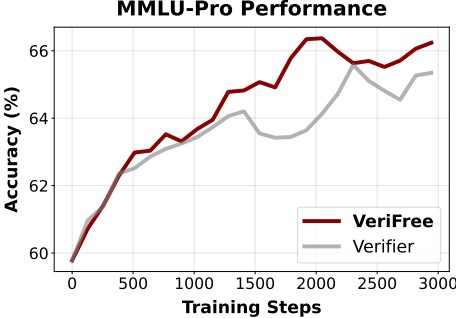 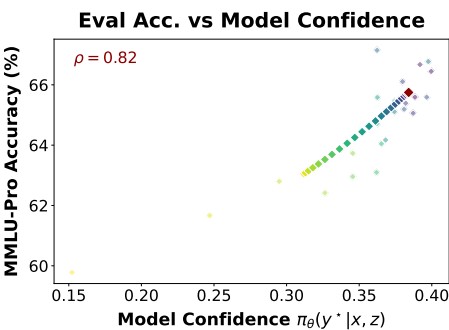

Figure 4: **Left**: MMLU-Pro accuracy of VeriFree and the baseline fine-tuned from Qwen3-8B base model along training steps. The curve is smoothed by a moving average with an interval of 384. **Right**: The dynamics of MMLU-Pro evaluation accuracy and average model confidence $\pi_\theta(\boldsymbol{y}^\star|\boldsymbol{x}, \boldsymbol{z})$ along training based on Qwen3-8B base model. Raw data points are depicted with more transparency, while smoothed data using a Gaussian filter is shown with less transparency for emphasis. Darker colors represent larger training steps.

tial gains in average accuracy (ranging from 12% to 40%), demonstrating that VeriFree effectively fine-tunes LLMs to improve general reasoning performance. See Appendix E for more results.

Similar improvements are observed on the SuperGPQA benchmark, as shown in Table 2, where VeriFree consistently enhances the performance of base models by a significant margin. Notably, VeriFree achieves performance comparable to, or even surpassing, that of the instruct model in *thinking* mode and the Verifier baseline (i.e., the RL-tuned model learned with an additional model-based verifier), without relying on any explicit verification signals.

In addition to accuracy gains, we also observe an increase in response length after tuned by VeriFree, suggesting that the model explores longer reasoning traces to arrive at more accurate answers, which is a behavior reminiscent of DeepSeek-R1-Zero (Guo et al., 2025). Results on the GPQA benchmark are provided in Appendix E due to space constraints.

**VeriFree leads to better learning efficiency.** We compare VeriFree with the baseline that learns from a model-based verifier reward (i.e., Verifier). As shown in Fig. 4 (Left), VeriFree consistently outperforms the baseline, achieving higher accuracy with fewer training steps. We attribute this improved learning efficiency to reduced gradient variance, enabled by VeriFree's continuous reward signals and the RLOO objective. While both approaches optimize the same reward signal in expectation, VeriFree provides more stable and informative policy gradients, which accelerate convergence and leads to better final performance.

**Model confidence is a good reasoning capability proxy.** Our analysis based on Qwen3-8B base model reveals a strong positive correlation ($\rho = 0.82$) between MMLU-Pro accuracy and the average model confidence $\pi_\theta(\boldsymbol{y}^\star|\boldsymbol{x}, \boldsymbol{z})$ during training (Fig. 4, Right). This empirically demonstrates that the model's self-estimated confidence in the correct answer, i.e., $\pi_\theta(\boldsymbol{y}^\star|\boldsymbol{x}, \boldsymbol{z})$, serves as an effective metric for quantifying emergent reasoning capabilities in LLMs.

**VeriFree learns transferable reasoning skills.** To evaluate the transferability of reasoning acquired through VeriFree, we train a model on a dataset with all math-related examples removed, and assess its performance on both general and math-specific benchmarks. As shown in Fig. 5, VeriFree not only improves reasoning performance on general tasks, as expected, but also demonstrates strong transfer to math benchmarks—despite the absence of math supervision during training. This highlights VeriFree's ability to induce general reasoning capabilities that extend across domains.

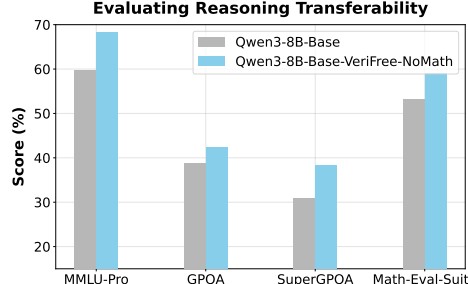

Figure 5: VeriFree enhances reasoning transfer to math without math training. When trained only on non-math data, the model improves on general benchmarks and effectively transfers to math-specific tasks.

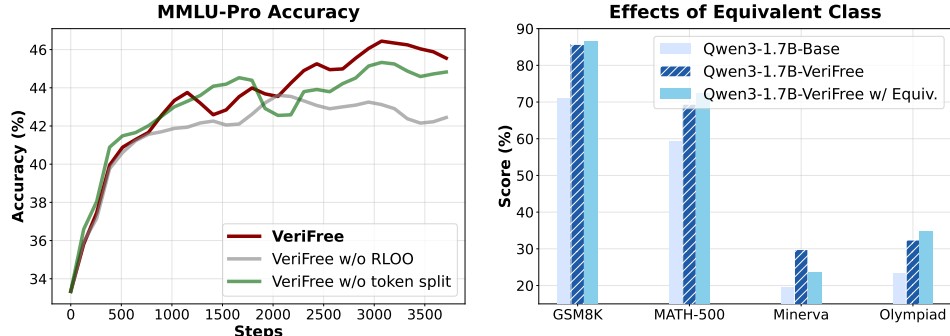

Figure 6: **Left**: MMLU-Pro evaluation accuracy of VeriFree against ablation variants (w/o RLOO, w/o tokenization-aware split strategy) along training steps. All models are based on Qwen3-1.7B base models. **Right**: Effects of introducing the equivalent class to VeriFree on model performance.

## 3.3 ABLATION STUDY

To systematically evaluate method components and offer a comprehensive understanding of VeriFree, we conduct ablation studies based on Qwen3-1.7B base models as follows.

**Effects of extraction strategy for reasoning trace $z$.** Our method requires precise separation between reasoning path $z$ and answer $y$ to enable answer replacement. While human-readable splits using `<answer>` seem intuitive, we instead define $z$ to end at "`<answer`" (omitting ">"), ensuring consistent tokenization boundaries (see Sec. 2.4). We compare with a variant using text-based splitting (denoted as "VeriFree w/o token split") on Qwen3-1.7B via MMLU-Pro (Fig. 6). Our tokenization-aware approach achieves superior convergence, while the variant suffers optimization instability due to effectively introducing off-policy data.

**Effects of RLOO.** As in Fig. 6 (Left), removing RLOO leads to a consistent drop in performance throughout training, with final accuracy more than $3\%$ lower than that of the full method. This highlights the importance of RLOO in stabilizing learning and guiding the model toward better generalization. Without RLOO, the model converges prematurely and fails to reach the same peak accuracy.

**Effects of equivalence class.** As mentioned in Sec. 2.1, verifier-based RL typically assesses answer correctness as rewards. Correct answers within a specific class often form an equivalence class. Our method, however, utilizes model confidence by focusing on a single reference answer for a given question and the model's reasoning trace. To explore the potential advantages of integrating an "equivalence class" into our approach, we conducted ablation studies as follows. We employed a model fine-tuned on the MATH-12k dataset (Hendrycks et al., 2021; Lightman et al., 2024) from Qwen3-8B base model through Dr. GRPO (Liu et al., 2025c) with rule-based verification to sample answers on MATH-12k, subsequently verifying answer correctness using Math-Verify. This approach enabled us to create an extended dataset with a set of equivalent correct answers for each question. We then fine-tuned Qwen3-1.7B base models using our method on both the original and the extended MATH-12k datasets incorporating equivalence classes. These models are evaluated on GSM8K, MATH-500, Minerva Math, and OlympiadBench to assess the impact of including equivalence classes. The results, shown in Fig. 6 (Right), indicate that considering equivalence classes offers slight performance improvements, aligning with our expectations. This highlights a minor limitation and motivates future work on algorithms that can better leverage answer equivalence.

## 4 CONCLUSIONS

In this paper, we rethink reinforcement learning with verifiable rewards (RLVR) for LLMs from a novel perspective. By leveraging the gradient equivalence under the unique answer assumption, we derive a new optimization objective that eliminates the need for explicit verification, whether rule-based or model-based. Our proposed method, VeriFree, is particularly well-suited for general reasoning tasks, where rule-based verifiers are infeasible and model-based verifiers are both expensive and vulnerable to reward hacking. Through extensive experiments and ablations, we demonstrate the effectiveness and robustness of VeriFree on a wide range of general reasoning benchmarks. We hope our work offers a fresh viewpoint for the LLM RL community and provides a practical approach for building future general-purpose reasoners.

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

# APPENDIX

## A    RELATED WORKS

**DeepSeek-R1-Zero-like reinforcement learning.** DeepSeek-R1-Zero (Guo et al., 2025), Tülu (Lambert et al., 2024) and OpenAI's o1 (OpenAI, 2024) recently demonstrated that applying RL to learn directly from binary verification-based rewards can be extremely powerful in enhancing the reasoning capabilities of base LLMs. Since then several works have reproduced R1-Zero-like training on smaller scales (Zeng et al., 2025; Pan et al., 2025; Liu et al., 2025b; Hu et al., 2025; Luo et al., 2025b;a). The aforementioned works all focus on math and coding, domains where verifiable rewards are readily available. By contrast, our work aims to extend the R1-Zero-like training paradigm to general domains where verifiable rewards are not available.

**Reasoning beyond verifiable domains.** Previous work on reasoning without access to verifiable rewards has been based around employing an additional LLM to act as a proxy verifier or reward model. NaturalReasoning (Yuan et al., 2025) introduces a large, multi-domain dataset and presents baselines trained using RFT (Yuan et al., 2023) and DPO (Rafailov et al., 2023), leveraging a second LLM as a reward model, while Su et al. (2025) and General-Reasoner (Ma et al., 2025) similarly incorporate a separate LLM to serve as a verifier. X-Reasoner (Liu et al., 2025a) also investigates general reasoning but circumvents the lack of rule-based verification by dropping the R1-Zero-style training paradigm, instead training via SFT (Grattafiori et al., 2024) on responses sampled from more capable models.

**Self-improving language models.** Several works have explored training LLMs using signals based on the model's own outputs. Yuan et al. (2024) propose to prompt the model to judge and rank different responses and select the best and worst for preference learning. Chen et al. (2025) leverage the DPO implicit rewards for more efficient and robust self-alignment via iterative DPO (Rafailov et al., 2023). Zuo et al. (2025) use majority voting to construct self-labeled rewards for RL to further improve well-trained models during test time, which can be understood as a form of sharpening (Huang et al., 2024). The aforementioned works differ from our approach. They do not require reference answers or labels, whereas our method requires reference answers in the training dataset. Furthermore, as shown in Sec. 2.2, it is proven that VeriFree is equivalent to RLVR under the single reference answer assumption. Another line of research (Phan et al., 2023; Chen et al., 2024b; Tang et al., 2025; Hu et al., 2024) approaches LLM reasoning from the direction of variational optimization, treating the reasoning trace as a latent variable. Despite starting from a different viewpoint our method has interesting and close connections to this perspective which we discuss in detail in Sec. 2.3.

## B    THEORETICAL ANALYSIS

### B.1    DERIVATION OF GRADIENT ESTIMATORS

Here we provide derivations of Eq. (5) for the gradient estimator of VeriFree. We also include the corresponding derivation for the standard verifier-based gradient estimator for completeness.

*Proof.* The gradient estimator for $J_{\text{Verifier}}$ is derived as follows:

$$\nabla_\theta J_{\text{Verifier}}(\theta; \boldsymbol{x}, \boldsymbol{y}^\star)$$

$$= \nabla_\theta \mathbb{E}_{\boldsymbol{z} \sim \pi_\theta(\cdot|\boldsymbol{x})} \Big[ \mathbb{E}_{\boldsymbol{y} \sim \pi_\theta(\cdot|\boldsymbol{x}, \boldsymbol{z})} \big[ R_{\text{Verifier}}(\boldsymbol{y}; \boldsymbol{y}^\star) \big] \Big]$$

$$= \nabla_\theta \sum_{\boldsymbol{z}, \boldsymbol{y}} R_{\text{Verifier}}(\boldsymbol{y}; \boldsymbol{y}^\star) \pi_\theta(\boldsymbol{y}|\boldsymbol{x}, \boldsymbol{z}) \pi_\theta(\boldsymbol{z}|\boldsymbol{x})$$

$$= \sum_{\boldsymbol{z}, \boldsymbol{y}} R_{\text{Verifier}}(\boldsymbol{y}; \boldsymbol{y}^\star) \big[ \pi_\theta(\boldsymbol{y}|\boldsymbol{x}, \boldsymbol{z}) \nabla_\theta \pi_\theta(\boldsymbol{z}|\boldsymbol{x}) + \pi_\theta(\boldsymbol{z}|\boldsymbol{x}) \nabla_\theta \pi_\theta(\boldsymbol{y}|\boldsymbol{x}, \boldsymbol{z}) \big]$$

$$= \sum_{\boldsymbol{z}, \boldsymbol{y}} R_{\text{Verifier}}(\boldsymbol{y}; \boldsymbol{y}^\star) \big[ \pi_\theta(\boldsymbol{y}|\boldsymbol{x}, \boldsymbol{z}) \pi_\theta(\boldsymbol{z}|\boldsymbol{x}) \nabla_\theta \log \pi_\theta(\boldsymbol{z}|\boldsymbol{x}) + \pi_\theta(\boldsymbol{z}|\boldsymbol{x}) \pi_\theta(\boldsymbol{y}|\boldsymbol{x}, \boldsymbol{z}) \nabla_\theta \log \pi_\theta(\boldsymbol{y}|\boldsymbol{x}, \boldsymbol{z}) \big]$$

$$= \mathbb{E}_{\boldsymbol{z} \sim \pi_\theta(\cdot|\boldsymbol{x})} \Big[ \mathbb{E}_{\boldsymbol{y} \sim \pi_\theta(\cdot|\boldsymbol{x}, \boldsymbol{z})} \big[ R_{\text{Verifier}}(\boldsymbol{y}; \boldsymbol{y}^\star) \big[ \nabla_\theta \log \pi_\theta(\boldsymbol{z}|\boldsymbol{x}) + \nabla_\theta \log \pi_\theta(\boldsymbol{y}|\boldsymbol{x}, \boldsymbol{z}) \big] \big] \Big]$$

The gradient estimator for $J_{\text{VeriFree}}$ is derived as follows:

$$\nabla_\theta J_{\text{VeriFree}}(\theta; \boldsymbol{x}, \boldsymbol{y}^\star)$$

$$= \nabla_\theta \mathbb{E}_{\boldsymbol{z} \sim \pi_\theta(\cdot|\boldsymbol{x})}[R_{\text{VeriFree}}(\boldsymbol{z}; \boldsymbol{x}, \boldsymbol{y}^\star)]$$

$$= \nabla_\theta \mathbb{E}_{\boldsymbol{z} \sim \pi_\theta(\cdot|\boldsymbol{x})}[\pi_\theta(\boldsymbol{y}^\star|\boldsymbol{x}, \boldsymbol{z})]$$

$$= \nabla_\theta \sum_{\boldsymbol{z}} \pi_\theta(\boldsymbol{y}^\star|\boldsymbol{x}, \boldsymbol{z})\pi_\theta(\boldsymbol{z}|\boldsymbol{x})$$

$$= \sum_{\boldsymbol{z}} \left[\pi_\theta(\boldsymbol{y}^\star|\boldsymbol{x}, \boldsymbol{z})\nabla_\theta \pi_\theta(\boldsymbol{z}|\boldsymbol{x}) + \pi_\theta(\boldsymbol{z}|\boldsymbol{x})\nabla_\theta \pi_\theta(\boldsymbol{y}^\star|\boldsymbol{x}, \boldsymbol{z})\right]$$

$$= \sum_{\boldsymbol{z}} \left[\pi_\theta(\boldsymbol{y}^\star|\boldsymbol{x}, \boldsymbol{z})\pi_\theta(\boldsymbol{z}|\boldsymbol{x})\nabla_\theta \log \pi_\theta(\boldsymbol{z}|\boldsymbol{x}) + \pi_\theta(\boldsymbol{z}|\boldsymbol{x})\pi_\theta(\boldsymbol{y}^\star|\boldsymbol{x}, \boldsymbol{z})\nabla_\theta \log \pi_\theta(\boldsymbol{y}^\star|\boldsymbol{x}, \boldsymbol{z})\right]$$

$$= \mathbb{E}_{\boldsymbol{z} \sim \pi_\theta(\cdot|\boldsymbol{x})}\left[\pi_\theta(\boldsymbol{y}^\star|\boldsymbol{x}, \boldsymbol{z})\left[\nabla_\theta \log \pi_\theta(\boldsymbol{z}|\boldsymbol{x}) + \nabla_\theta \log \pi_\theta(\boldsymbol{y}^\star|\boldsymbol{x}, \boldsymbol{z})\right]\right]$$

$$= \mathbb{E}_{\boldsymbol{z} \sim \pi_\theta(\cdot|\boldsymbol{x})}\left[R_{\text{VeriFree}}(\boldsymbol{z}; \boldsymbol{x}, \boldsymbol{y}^\star)\left[\nabla_\theta \log \pi_\theta(\boldsymbol{z}|\boldsymbol{x}) + \nabla_\theta \log \pi_\theta(\boldsymbol{y}^\star|\boldsymbol{x}, \boldsymbol{z})\right]\right]$$

$\square$

## B.2 PROOF OF LOWER VARIANCE

Here we provide a full proof of Theorem 1, the reduced variance property of VeriFree. We show that the policy gradient estimator derived from $J_{\text{VeriFree}}(\theta; \boldsymbol{x}, \boldsymbol{y}^\star)$ has variance less than or equal to that of the estimator derived from $J_{\text{Verifier}}(\theta; \boldsymbol{x}, \boldsymbol{y}^\star)$ for any given $\boldsymbol{x}, \boldsymbol{y}^\star$. The same relationship will hold for the global objectives averaged over $(\boldsymbol{x}, \boldsymbol{y}^\star) \sim \mathcal{D}$.

*Proof.* The global objective functions are:

$$J_{\text{Verifier}}(\theta) = \mathbb{E}_{(\boldsymbol{x}, \boldsymbol{y}^\star) \sim \mathcal{D}}\left[\mathbb{E}_{\boldsymbol{z} \sim \pi_\theta(\cdot|\boldsymbol{x})}\left[\mathbb{E}_{\boldsymbol{y} \sim \pi_\theta(\cdot|\boldsymbol{x}, \boldsymbol{z})}\left[\mathbb{1}\{\boldsymbol{y} = \boldsymbol{y}^\star\}\right]\right]\right]$$

$$J_{\text{VeriFree}}(\theta) = \mathbb{E}_{(\boldsymbol{x}, \boldsymbol{y}^\star) \sim \mathcal{D}}\left[\mathbb{E}_{\boldsymbol{z} \sim \pi_\theta(\cdot|\boldsymbol{x})}\left[\pi_\theta(\boldsymbol{y}^\star|\boldsymbol{x}, \boldsymbol{z})\right]\right]$$

For a given $(\boldsymbol{x}, \boldsymbol{y}^\star) \sim \mathcal{D}$, the single-sample Monte Carlo gradient estimators are:

$$\hat{G}_{\text{Verifier}}(\boldsymbol{x}, \boldsymbol{y}^\star, \boldsymbol{z}, \boldsymbol{y}) = \mathbb{1}\{\boldsymbol{y} = \boldsymbol{y}^\star\}\left[\nabla_\theta \log \pi_\theta(\boldsymbol{z}|\boldsymbol{x}) + \nabla_\theta \log \pi_\theta(\boldsymbol{y}|\boldsymbol{x}, \boldsymbol{z})\right]$$

where $\boldsymbol{z} \sim \pi_\theta(\cdot|\boldsymbol{x})$, $\boldsymbol{y} \sim \pi_\theta(\cdot|\boldsymbol{x}, \boldsymbol{z})$, and

$$\hat{G}_{\text{VeriFree}}(\boldsymbol{x}, \boldsymbol{y}^\star, \boldsymbol{z}) = \pi_\theta(\boldsymbol{y}^\star|\boldsymbol{x}, \boldsymbol{z})\left[\nabla_\theta \log \pi_\theta(\boldsymbol{z}|\boldsymbol{x}) + \nabla_\theta \log \pi_\theta(\boldsymbol{y}^\star|\boldsymbol{x}, \boldsymbol{z})\right]$$

where $\boldsymbol{z} \sim \pi_\theta(\cdot|\boldsymbol{x})$.

The proof relies on the law of total variance and the relationship between $\hat{G}_{\text{Verifier}}$ and $\hat{G}_{\text{VeriFree}}$.

First, we show that $\hat{G}_{\text{VeriFree}}(\boldsymbol{x}, \boldsymbol{y}^\star, \boldsymbol{z})$ is the conditional expectation of $\hat{G}_{\text{Verifier}}(\boldsymbol{x}, \boldsymbol{y}^\star, \boldsymbol{z}, \boldsymbol{y})$ given $\boldsymbol{x}, \boldsymbol{y}^\star, \boldsymbol{z}$. The expectation is taken over $\boldsymbol{y} \sim \pi_\theta(\cdot|\boldsymbol{x}, \boldsymbol{z})$:

$$\mathbb{E}_{\boldsymbol{y} \sim \pi_\theta(\cdot|\boldsymbol{x}, \boldsymbol{z})}\left[\hat{G}_{\text{Verifier}}(\boldsymbol{x}, \boldsymbol{y}^\star, \boldsymbol{z}, \boldsymbol{y})|\boldsymbol{x}, \boldsymbol{y}^\star, \boldsymbol{z}\right]$$

$$= \mathbb{E}_{\boldsymbol{y} \sim \pi_\theta(\cdot|\boldsymbol{x}, \boldsymbol{z})}\left[\mathbb{1}\{\boldsymbol{y} = \boldsymbol{y}^\star\}\left[\nabla_\theta \log \pi_\theta(\boldsymbol{z}|\boldsymbol{x}) + \nabla_\theta \log \pi_\theta(\boldsymbol{y}|\boldsymbol{x}, \boldsymbol{z})\right]\Big|\boldsymbol{x}, \boldsymbol{y}^\star, \boldsymbol{z}\right]$$

$$= \sum_{\boldsymbol{y}'} \pi_\theta(\boldsymbol{y}'|\boldsymbol{x}, \boldsymbol{z})\left[\mathbb{1}\{\boldsymbol{y}' = \boldsymbol{y}^\star\}\left[\nabla_\theta \log \pi_\theta(\boldsymbol{z}|\boldsymbol{x}) + \nabla_\theta \log \pi_\theta(\boldsymbol{y}'|\boldsymbol{x}, \boldsymbol{z})\right]\right]$$

$$= \pi_\theta(\boldsymbol{y}^\star|\boldsymbol{x}, \boldsymbol{z})\left[\nabla_\theta \log \pi_\theta(\boldsymbol{z}|\boldsymbol{x}) + \nabla_\theta \log \pi_\theta(\boldsymbol{y}^\star|\boldsymbol{x}, \boldsymbol{z})\right]$$

$$= \hat{G}_{\text{VeriFree}}(\boldsymbol{x}, \boldsymbol{y}^\star, \boldsymbol{z})$$

We denote $\mathbb{E}_{\boldsymbol{y} \sim \pi_\theta(\cdot|\boldsymbol{x},\boldsymbol{z})}\big[\hat{G}_{\text{Verifier}}(\boldsymbol{x}, \boldsymbol{y}^\star, \boldsymbol{z}, \boldsymbol{y})\big|\boldsymbol{x}, \boldsymbol{y}^\star, \boldsymbol{z}\big]$ as $\mathbb{E}_{\boldsymbol{y}|\boldsymbol{z}}\big[\hat{G}_{\text{Verifier}}(\boldsymbol{x}, \boldsymbol{y}^\star, \boldsymbol{z}, \boldsymbol{y})\big]$ for brevity in the following, since $\boldsymbol{x}$ and $\boldsymbol{y}^\star$ are already given and fixed. Thus, we have

$$\mathbb{E}_{\boldsymbol{y}|\boldsymbol{z}}\big[\hat{G}_{\text{Verifier}}(\boldsymbol{x}, \boldsymbol{y}^\star, \boldsymbol{z}, \boldsymbol{y})\big] = \hat{G}_{\text{VeriFree}}(\boldsymbol{x}, \boldsymbol{y}^\star, \boldsymbol{z}) \tag{8}$$

The law of total variance states that for a random variable $W$ and conditioning variables $S$, $\text{Var}(W) = \mathbb{E}_S[\text{Var}[W|S]] + \text{Var}_S[\mathbb{E}[W|S]]$. Let $W = \hat{G}_{\text{Verifier}}(\boldsymbol{x}, \boldsymbol{y}^\star, \boldsymbol{z}, \boldsymbol{y})$. Given $\boldsymbol{x}$ and $\boldsymbol{y}^\star$, the sources of randomness for $\hat{G}_{\text{Verifier}}$ are $\boldsymbol{z}$ and $\boldsymbol{y}$. Let $S = \boldsymbol{z}$ be the conditioning variables. The randomness in $\hat{G}_{\text{Verifier}}$ given $S$ comes from $\boldsymbol{y} \sim \pi_\theta(\cdot|\boldsymbol{x}, \boldsymbol{z})$. Applying the law:

$$\text{Var}_{\boldsymbol{z},\boldsymbol{y}}\big[\hat{G}_{\text{Verifier}}(\boldsymbol{x}, \boldsymbol{y}^\star, \boldsymbol{z}, \boldsymbol{y})\big] = \mathbb{E}_{\boldsymbol{z}}\Big[\text{Var}_{\boldsymbol{y}|\boldsymbol{z}}\big[\hat{G}_{\text{Verifier}}(\boldsymbol{x}, \boldsymbol{y}^\star, \boldsymbol{z}, \boldsymbol{y})\big]\Big] + \text{Var}_{\boldsymbol{z}}\Big[\mathbb{E}_{\boldsymbol{y}|\boldsymbol{z}}\big[\hat{G}_{\text{Verifier}}(\boldsymbol{x}, \boldsymbol{y}^\star, \boldsymbol{z}, \boldsymbol{y})\big]\Big]$$

The expectation $\mathbb{E}_{\boldsymbol{z}}$ is over $\boldsymbol{z} \sim \pi_\theta(\cdot|\boldsymbol{x})$. The conditional variance $\text{Var}_{\boldsymbol{y}|\boldsymbol{z}}$ and expectation $\mathbb{E}_{\boldsymbol{y}|\boldsymbol{z}}$ are over $\boldsymbol{y} \sim \pi_\theta(\cdot|\boldsymbol{x}, \boldsymbol{z})$ for fixed $\boldsymbol{x}, \boldsymbol{y}^\star, \boldsymbol{z}$.

Substituting the result from Eq. (8) into the law of total variance:

$$\text{Var}_{\boldsymbol{z},\boldsymbol{y}}\big[\hat{G}_{\text{Verifier}}(\boldsymbol{x}, \boldsymbol{y}^\star, \boldsymbol{z}, \boldsymbol{y})\big] = \mathbb{E}_{\boldsymbol{z}}\Big[\text{Var}_{\boldsymbol{y}|\boldsymbol{z}}\big[\hat{G}_{\text{Verifier}}(\boldsymbol{x}, \boldsymbol{y}^\star, \boldsymbol{z}, \boldsymbol{y})\big]\Big] + \text{Var}_{\boldsymbol{z}}\big[\hat{G}_{\text{VeriFree}}(\boldsymbol{x}, \boldsymbol{y}^\star, \boldsymbol{z})\big]$$

The second term, $\text{Var}_{\boldsymbol{z}}\big[\hat{G}_{\text{VeriFree}}(\boldsymbol{x}, \boldsymbol{y}^\star, \boldsymbol{z})\big]$, is the definition of the variance of the estimator $\hat{G}_{\text{VeriFree}}$. The first term, $\mathbb{E}_{\boldsymbol{z}}\Big[\text{Var}_{\boldsymbol{y}|\boldsymbol{z}}\big[\hat{G}_{\text{Verifier}}(\boldsymbol{x}, \boldsymbol{y}^\star, \boldsymbol{z})\big]\Big]$, is an expectation of a variance. Since variance is always non-negative, $\text{Var}_{\boldsymbol{y}|\boldsymbol{z}}\big[\hat{G}_{\text{Verifier}}(\boldsymbol{x}, \boldsymbol{y}^\star, \boldsymbol{z})\big] \geq 0$. Therefore, its expectation is also non-negative:

$$\mathbb{E}_{\boldsymbol{z}}\Big[\text{Var}_{\boldsymbol{y}|\boldsymbol{z}}\big[\hat{G}_{\text{Verifier}}(\boldsymbol{x}, \boldsymbol{y}^\star, \boldsymbol{z})\big]\Big] \geq 0$$

Thus, we have:

$$\text{Var}_{\boldsymbol{z},\boldsymbol{y}}\big[\hat{G}_{\text{Verifier}}(\boldsymbol{x}, \boldsymbol{y}^\star, \boldsymbol{z}, \boldsymbol{y})\big] = (\text{a non-negative term}) + \text{Var}_{\boldsymbol{z}}\big[\hat{G}_{\text{VeriFree}}(\boldsymbol{x}, \boldsymbol{y}^\star, \boldsymbol{z})\big]$$

This implies:

$$\text{Var}_{\boldsymbol{z},\boldsymbol{y}}\big[\hat{G}_{\text{Verifier}}(\boldsymbol{x}, \boldsymbol{y}^\star, \boldsymbol{z}, \boldsymbol{y})\big] \geq \text{Var}_{\boldsymbol{z}}\big[\hat{G}_{\text{VeriFree}}(\boldsymbol{x}, \boldsymbol{y}^\star, \boldsymbol{z})\big]$$

The variance of the policy gradient estimator $\hat{G}_{\text{VeriFree}}$ is less than or equal to that of $\hat{G}_{\text{Verifier}}$. This is an instance of Rao-Blackwellization, where analytically integrating out a source of randomness (the sampling of $\boldsymbol{y}$) by using its conditional expectation reduces variance. $\qquad\square$

## C  OFF-POLICY GRADIENT ESTIMATORS

In the main paper we provide an expression for the gradient estimator when the data is fully on-policy. VeriFree is also fully compatible with PPO-style gradient clipping for the case when data is reused to improve sample efficiency. In this case the gradient estimator is:

$$\nabla_\theta J_{\text{VeriFree}}(\theta) = \frac{1}{G} \sum_{i=1}^{G} \Bigg[ \sum_{t=1}^{|\boldsymbol{z}_i|} \text{Clip}\bigg\{ \frac{\pi_\theta(\boldsymbol{z}_{i,t} \mid \boldsymbol{x}, \boldsymbol{z}_{i,<t})}{\pi_{\theta_{\text{old}}}(\boldsymbol{z}_{i,t} \mid \boldsymbol{x}, \boldsymbol{z}_{i,<t})} \bigg\} A_i + \sum_{t'=1}^{|\boldsymbol{y}^\star|} \text{Clip}\bigg\{ \frac{\pi_\theta(\boldsymbol{y}_t^\star \mid \boldsymbol{x}, \boldsymbol{z}_i)}{\pi_{\theta_{\text{old}}}(\boldsymbol{y}_t^\star \mid \boldsymbol{x}, \boldsymbol{z}_i)} \bigg\} R_i \Bigg],$$

where $\pi_{\theta_{\text{old}}}$ is the sampling policy, $A_i = \pi_{\theta_{\text{old}}}(\boldsymbol{y}^\star|\boldsymbol{x}, \boldsymbol{z}_i) - \frac{1}{G-1}\sum_{j \neq i} \pi_{\theta_{\text{old}}}(\boldsymbol{y}^\star|\boldsymbol{x}, \boldsymbol{z}_j)$, $R_i = \pi_{\theta_{\text{old}}}(\boldsymbol{y}^\star|\boldsymbol{x}, \boldsymbol{z}_i)$, and $\text{Clip}\{\cdot\}$ denotes the PPO clipping operation.

## D  DATASET DETAILS

The category distribution in WebData (our training data) is visualized in Fig. 7.

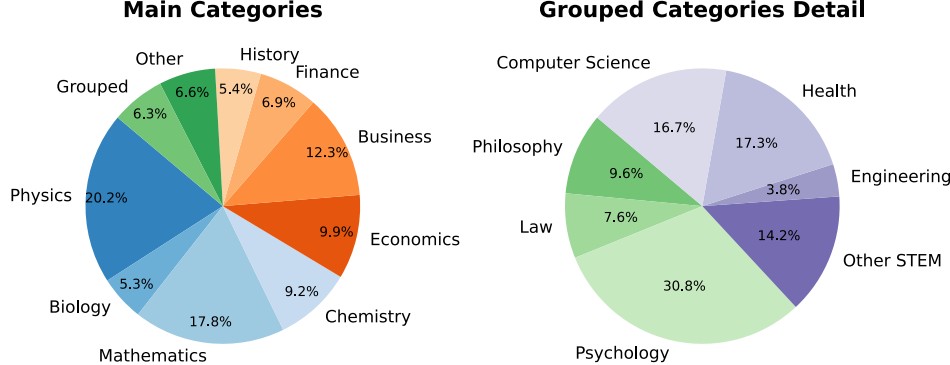

Figure 7: Category distributions in WebData. A breakdown of the "grouped" category (right) shows detailed distribution of various domains with fewer data samples.

# E  EXTENDED EMPIRICAL RESULTS

## E.1  MATH-RELATED BENCHMARKS AND GPQA-DIAMOND

In Tables 1 and 2, we provide detailed benchmark results for MMLU-Pro and SuperGPQA using domain name abbreviations. The full nomenclature is as follows:

**MMLU-Pro** (Table 1):
CS (Computer Science), Math (Mathematics), Chem (Chemistry), Eng (Engineering),
Law (Law), Bio (Biology), Health (Health), Phys (Physics), Bus (Business),
Phil (Philosophy), Econ (Economics), Other (Other), Psy (Psychology), Hist (History).

**SuperGPQA** (Table 2):
Eng. (Engineering), Med. (Medicine), Sci. (Science), Phil. (Philosophy),
M.S. (Military Science), Econ. (Economics), Mgmt. (Management), Socio. (Sociology),
L/A (Literature and Arts), Hist. (History), Agron. (Agronomy), Law (Law), Edu (Education).

In Table 3, we present an accuracy comparison across six math evaluation benchmarks and GPQA-Diamond. Models trained with VeriFree demonstrate consistent and significant improvements over the base models, further validating the effectiveness of our approach.

Table 3: Accuracy comparison on math evaluation suite and GPQA-Diamond.

| Method | AIME24 | | AMC | | GSM8K | | MATH-500 | | Minerva | | Olympiad | | GPQA-D | |
|---|---|---|---|---|---|---|---|---|---|---|---|---|---|---|
| | Acc. | Len. | Acc. | Len. | Acc. | Len. | Acc. | Len. | Acc. | Len. | Acc. | Len. | Acc. | Len. |
| Qwen3-1.7B-Base | 1.7 | 1287 | 40.0 | 1001 | 71.4 | 343 | 58.8 | 1094 | 19.5 | 1071 | 23.3 | 1737 | 17.7 | 1355 |
| Qwen3-1.7B (w/o thinking) | 10.9 | 2552 | 40.0 | 2291 | 83.3 | 312 | 72.8 | 1021 | 27.9 | 776 | 39.1 | 1970 | 19.2 | 1885 |
| Qwen3-1.7B (w/ thinking) | 20.9 | 7855 | 57.5 | 5973 | 89.0 | 2220 | 77.4 | 4525 | 36.8 | 5606 | 40.9 | 6497 | 17.7 | 7104 |
| Qwen3-1.7B-Base-Verifier | 8.4 | 2317 | 42.5 | 1712 | 81.7 | 405 | 66.2 | 1057 | 30.5 | 1357 | 29.8 | 1885 | 36.4 | 1317 |
| Qwen3-1.7B-Base-VeriFree | 10.7 | 1783 | 37.5 | 1522 | 76.2 | 447 | 63.8 | 972 | 21.0 | 867 | 29.8 | 1484 | 30.3 | 1147 |
| Qwen3-4B-Base | 5.2 | 1312 | 50.0 | 1001 | 73.1 | 393 | 73.4 | 724 | 29.4 | 1027 | 39.6 | 1202 | 24.7 | 1275 |
| Qwen3-4B (w/o thinking) | 20.4 | 3146 | 67.5 | 2047 | 92.1 | 308 | 82.2 | 1143 | 41.2 | 822 | 49.5 | 2593 | 29.8 | 2127 |
| Qwen3-4B (w/ thinking) | 33.0 | 7750 | 62.5 | 6123 | 92.9 | 2261 | 84.4 | 4370 | 41.5 | 5352 | 47.3 | 6432 | 31.8 | 6504 |
| Qwen3-4B-Base-Verifier | 15.6 | 2407 | 57.5 | 1566 | 72.5 | 377 | 73.8 | 1023 | 24.6 | 1088 | 45.9 | 1576 | 44.4 | 1266 |
| Qwen3-4B-Base-VeriFree | 16.9 | 2706 | 65.0 | 1904 | 87.5 | 682 | 74.8 | 1269 | 25.4 | 1444 | 44.9 | 1899 | 42.4 | 1619 |
| Qwen2.5-7B | 2.2 | 869 | 32.5 | 922 | 84.6 | 260 | 63.2 | 582 | 26.8 | 819 | 30.2 | 963 | 24.2 | 595 |
| Qwen-2.5-7B-SimpleRL-Zoo | 15.5 | 1285 | 57.5 | 1097 | 92.1 | 331 | 78.6 | 690 | 36.4 | 795 | 43.4 | 1083 | 23.7 | 990 |
| Qwen2.5-Math-7B-Oat-Zero | 28.3 | 1115 | 65.0 | 846 | 90.8 | 386 | 79.0 | 652 | 33.1 | 655 | 43.0 | 860 | 24.7 | 722 |
| Qwen2.5-7B-Instruct | 11.2 | 993 | 52.5 | 986 | 91.7 | 318 | 78.2 | 649 | 37.9 | 690 | 39.9 | 1123 | 31.8 | 643 |
| General-Reasoner-7B | 13.1 | 1363 | 52.5 | 1083 | 81.7 | 408 | 74.6 | 858 | 23.5 | 968 | 39.3 | 1303 | 34.8 | 1199 |
| Qwen3-8B-Base | 6.5 | 1213 | 65.0 | 917 | 91.7 | 304 | 77.0 | 671 | 38.2 | 884 | 41.3 | 1189 | 38.9 | 887 |
| Qwen3-8B (w/o thinking) | 24.6 | 2897 | 62.5 | 1919 | 93.3 | 293 | 82.2 | 1087 | 36.4 | 751 | 49.5 | 2422 | 27.8 | 2204 |
| Qwen3-8B (w/ thinking) | 31.0 | 7789 | 62.5 | 6201 | 95.3 | 2203 | 83.0 | 4414 | 43.8 | 5410 | 45.2 | 6554 | 37.4 | 6681 |
| Qwen3-8B-Base-Verifier | 17.8 | 1954 | 57.5 | 1265 | 93.4 | 307 | 76.2 | 784 | 39.0 | 698 | 41.8 | 1201 | 43.9 | 733 |
| Qwen3-8B-Base-VeriFree | 25.2 | 2461 | 67.5 | 1542 | 91.6 | 375 | 80.8 | 1024 | 30.5 | 902 | 50.2 | 1778 | 44.4 | 1080 |

## E.2 EMPIRICAL COMPARISON WITH RELATED APPROACHES

In Sec. 2.3, we have discussed the methodological differences between our work and related approaches, LaTRO and JEPO. In this section, we present an experimental comparison with these methods.

We first evaluate JEPO as a new baseline by fine-tuning the Qwen3 base model on our curated dataset (WebData) as introduced in Sec. 3.1. During fine-tuning, we observe that the number of reasoning tokens in the model's responses progressively decreases, eventually leaving almost no valid reasoning tokens (only answer tokens). The resulting performance is worse than that of the base model, so we do not report these results here. A similar trend is observed for LaTRO. We hypothesize that this degradation stems from two factors: (1) potential noise in the training data (which is preprocessed from Web Instruct and filtered by an LLM), and (2) the lack of robustness in LaTRO/JEPO, as these methods apply uniform weighting to answer token gradients. This may force the model to prioritize reference answers even when the reasoning quality is poor (see Sec. 2.3 for further analysis).

To further investigate, we evaluate JEPO and our method on the cleaner math12k dataset (Hendrycks et al., 2021) for mathematical reasoning tasks. We fine-tune the Qwen3-4B-Base model using four approaches: (1) RL with rule-based verification (denoted as RLVR), (2) LaTRO, (3) JEPO, and (4) VeriFree. Evaluation across multiple mathematical reasoning benchmarks (see Table 4) shows that our method consistently outperforms JEPO and matches the performance of RL with rule-based verification.

Table 4: Accuracy comparison on math tasks. All methods are **trained on math12k** (Hendrycks et al., 2021) based on **Qwen3-4B-Base**. The best results are highlighted using **bold text**.

| Method | MATH500 | AIME24 | AMC | Minerva | Olympiad | GSM8K |
|---|---|---|---|---|---|---|
| Qwen3-4B-Base | 73.4 | 5.2 | 50 | 29.4 | 38.6 | 73.1 |
| RLVR | 83.8 | 18.54 | **62.5** | **42.65** | 50.07 | **92.7** |
| LaTRO | 82.6 | 16.67 | 60.0 | 37.9 | 45.9 | 90.8 |
| JLB | 83 | 16.67 | 60.0 | 39.0 | 47.7 | 92.0 |
| VeriFree | **84.6** | **21.46** | **62.5** | **42.65** | **50.22** | 92.57 |

For a direct comparison with General Reasoner (Ma et al., 2025) which also focuses on general domain reasoning, we evaluate their newly released General-Reasoner-4B (also fine-tuned from Qwen3-4B-Base) on MMLU-Pro, SuperGPQA, and GPQA-Diamond. These benchmarks are designed to assess reasoning capabilities across diverse general domains. The results, presented in Table 5, demonstrate that our method matches or surpasses other approaches of a comparable model scale.

Table 5: Accuracy comparison on general domain reasoning tasks. All methods are based on **Qwen3-4B-Base**. The best results are highlighted using **bold text**. Note that [†] indicates results reported by Ma et al. (2025).

| Model | MMLU-Pro | SuperGPQA | GPQA-D |
|---|---|---|---|
| Qwen3-4B-Base[†] | 51.6 | 25.4 | 26.3 |
| Qwen3-4B (non-think)[†] | 61.8 | 32.1 | 41.7 |
| General-Reasoner-4B | 62.8 | 32.5 | 42.9 |
| Qwen3-4B-Base-Verifier | 63.0 | 34.3 | **44.4** |
| Qwen3-4B-Base-VeriFree | **63.5** | **35.1** | 42.4 |

## E.3 EVALUATION RESULTS WITH DIFFERENT SAMPLING PARAMETERS

The sampling parameters used in our main experiment (detailed in Sec. 3.1) follow the configuration from Ma et al. (2025). Recent works (Hochlehnert et al., 2025; Team, 2025) suggest that greedy decoding may be suboptimal for reasoning models that generate extended thinking traces. To address

this concern, we conduct a comprehensive evaluation on GPQA-Diamond using various sampling configurations.

We provide additional evaluation results on GPQA-Diamond using six distinct sampling configurations, including those recommended in the Qwen3 official report (Team, 2025):

- S1: temperature=0 (i.e., greedy decoding) and 8k token budget (same as Ma et al. (2025))
- S2: temperature=0 (i.e., greedy decoding) and 32k token budget
- S3: temperature=0.7, top_p=0.8, top_k=20, min_p=0, n_repeats=10, and 8k token budget
- S4: temperature=0.7, top_p=0.8, top_k=20, min_p=0, n_repeats=10, and 32k token budget
- S5: temperature=0.6, top_p=0.95, top_k=20, min_p=0, n_repeats=10, and 8k token budget
- S6: temperature=0.6, top_p=0.95, top_k=20, min_p=0, n_repeats=10, and 32k token budget

We report the results in Table 6. As the results show, our method consistently matches or surpasses both the counterpart with the model-based verifier and also the Qwen (w/o thinking) across most settings. Qwen3 (w/ thinking) shows strong results when given extremely large token budgets (32k). Note that Qwen3 (w/ thinking) is trained with multiple stages, including cold start with distillation data, while our method and the counterparts with verifiers are directly trained from the base model ("zero" setting).

Table 6: Performance comparison with different sampling configurations on GPQA-Diamond. The best results are highlighted using **bold text**.

| Model | S1 | S2 | S3 | S4 | S5 | S6 |
|---|---|---|---|---|---|---|
| Qwen3-4B-Base | 28.79 | 28.28 | 29.85 | 29.39 | 29.49 | 28.99 |
| Qwen3-4B (w/o thinking) | 36.87 | 31.31 | 38.99 | 40.05 | 37.88 | 38.43 |
| Qwen3-4B (w/ thinking) | 29.29 | 47.47 | 36.26 | **55.15** | 35.66 | **54.19** |
| Qwen3-4B-Base-Verifier | **42.42** | 43.43 | 41.57 | 42.37 | 42.07 | 41.41 |
| Qwen3-4B-Base-VeriFree | 40.91 | **45.45** | **43.03** | 42.88 | **43.99** | 42.88 |
| Qwen3-8B-Base | 37.37 | 37.88 | 34.75 | 36.87 | 36.62 | 34.85 |
| Qwen3-8B (w/o thinking) | 35.86 | 32.32 | 37.02 | 36.46 | 36.41 | 37.83 |
| Qwen3-8B (w/ thinking) | 36.36 | **58.59** | 37.37 | **61.26** | 37.83 | **62.27** |
| Qwen3-8B-Base-Verifier | 40.91 | 44.95 | 46.52 | 47.27 | 45.76 | 47.68 |
| Qwen3-8B-Base-VeriFree | **48.48** | 43.94 | **47.22** | 46.46 | **46.01** | 47.37 |

### E.4 Avg@K performance on Small Benchmarks

We note that the primary benchmarks in our evaluation are large in scale. For instance, MMLU-Pro includes 12k test samples and SuperGPQA contains 26k. On these large benchmarks, VeriFree generally achieves stronger performance.

We have also added evaluations on smaller benchmarks, as shown in Table 7. We report Avg@K, where K is chosen such that the total number of samples after repetition is approximately 1k.

Table 7: Avg@K on small benchmarks. The best results are highlighted using **bold text**.

| Model | GPQA-D Avg@4 | AIME24 Avg@32 | MATH500 Avg@2 | Minerva Avg@4 |
|---|---|---|---|---|
| Qwen3-4B-Base-Verifier | **42.80** | 16.04 | 73.70 | 24.63 |
| Qwen3-4B-Base-VeriFree | 42.29 | **17.50** | **75.10** | **25.54** |
| Qwen3-8B-Base-Verifier | 44.07 | 18.96 | 77.20 | **36.50** |
| Qwen3-8B-Base-VeriFree | **45.45** | **25.52** | **81.10** | 34.67 |

## E.5 EXPERIMENTS ON OTHER MODEL FAMILIES

We have extended our evaluation to an additional model family, applying both our method and the verifier-based baseline to OctoThinker-3B-Long-Base (Wang et al., 2025), a model mid-trained from Llama-3. The performance is compared against the official RLVR-trained reasoning model (OctoThinker-3B-Long-Zero) and the corresponding verifier-based baseline. All evaluations are conducted under a zero-shot setting. Results are presented in Table 8. Our model can outperform both the official RLVR model and our implemented counterpart with a model-based verifier. The base model does not perform well due to poor instruction-following abilities.

Table 8: Accuracy comparison on general domain reasoning tasks. All methods are based on **OctoThinker-3B-Long-Base**. The best results are highlighted using **bold text**.

| Model | MMLU-Pro | SuperGPQA | GPQA-D |
|---|---|---|---|
| OctoThinker-3B-Long-Base | 5.2 | 2.1 | 0.0 |
| OctoThinker-3B-Long-Zero | 35.8 | 18.6 | 25.3 |
| OctoThinker-3B-Long-Verifier | 35.5 | 19.8 | 28.3 |
| OctoThinker-3B-Long-VeriFree | **36.2** | **20.5** | **30.3** |

## E.6 EXPERIMENTS WITH LONGER ANSWERS

In our main experiments, the dataset preprocessing retains only short-phrase reference answers with token lengths under 7. To investigate whether VeriFree can effectively handle longer answers, we conduct an additional experiment using the training data from General Reasoner (Ma et al., 2025), which includes examples with longer reference answers. We align the training and evaluation protocols with General Reasoner, and denote this variant as "Qwen3-4B-Base-VeriFree w/ GR data". The results are presented in Table 9.

Table 9: Experiments with longer reference answers. All methods are based on **Qwen3-4B-Base**. The best results are highlighted using **bold text**.

| Model | GPQA-D | MMLU-Pro | SuperGPQA |
|---|---|---|---|
| Qwen3-4B-Base-Verifier | **44.4** | 63.0 | 34.3 |
| Qwen3-4B-Base-VeriFree w/ our data | 42.4 | **63.5** | **35.1** |
| General-Reasoner-4B | 42.9 | 62.8 | 32.5 |
| Qwen3-4B-Base-VeriFree w/ GR data | **45.5** | **63.2** | **34.5** |

Even when trained on data containing longer reference answers, VeriFree continues to outperform the baselines. It is also noteworthy that our method remains more computationally efficient.

## F DISCUSSION

This section provides a detailed discussion of the advantages and limitations of our proposed method, VeriFree.

## F.1 ADVANTAGES

VeriFree offers distinct advantages over other verification paradigms for general LLM reasoning tasks. While rule-based verification is effective, its applicability is restricted to specialized domains like mathematics and coding. In contrast, model-based verifiers (e.g., LLM-as-judge methods) offer broader applicability but introduce practical challenges, including substantial resource overhead (e.g., additional training data, increased computation, and memory usage) and inherent risks such as reward hacking and various biases (Ye et al., 2025; Chen et al., 2024a). A prominent example is length bias, where LLMs systematically favor longer responses irrespective of quality. The fact that methods like General Reasoner (Ma et al., 2025) must explicitly incorporate length-based penalties

(Sec. 3.1) underscores the pervasiveness of this issue. VeriFree circumvents these issues by directly optimizing the conditional likelihood of the reference answer, eliminating the need for an auxiliary verifier model. So it avoids these pitfalls entirely, providing a more robust and efficient alternative.

Although VeriFree is trained using only a single reference answer, it remains highly effective even when multiple correct solutions exist. This is because the pretrained base model learns a generalized understanding of correctness and reasoning logic, rather than merely memorizing a specific answer format. Much like in supervised fine-tuning, where models develop broad instruction-following ability despite being trained on fixed-style examples, VeriFree captures the underlying principles of valid reasoning, enabling it to recognize correctness across varied expressions. This demonstrates VeriFree's superiority in generalization without relying on multiple references during training.

## F.2 LIMITATIONS

The primary limitation of VeriFree is its dependency on direct question-answer pairs for supervision (note that VeriFree does not require any reference reasoning traces). This requirement constrains its applicability in scenarios where such supervised pairs are unavailable. For instance, in certain coding tasks, only test cases may be provided without corresponding reference implementations. In such settings, rule-based verification or other methods that do not rely on reference answers may represent the only viable alternatives.

## G CASE STUDY

In this section, we provide some examples of reasoning traces produced by the model trained by our method on training questions. And we also provide an example with both our model's and base model's responses on the test dataset.

---

**Example question in training dataset (category: Chemistry)**

**Prompt:**

```
<|im\_start|>user
Label or a placard colored black and white with a skull and
crossbones falls under which class
Please reason step by step, and put your final answer within
<answer> \boxed{} </answer>.<|im\_end|>
<|im\_start|>assistant
```

**Our Model's Response:**

```
A label or placard colored black and white with a skull and
crossbones indicates the class of hazardous materials known as
**Class 6**. Class 6 is specifically divided into two divisions:

- **Division 6.1**: Toxic substances. These are materials that can
cause injury or death if ingested or inhaled.
- **Division 6.2**: Infectious substances. These are materials
that contain pathogens and can cause disease in humans or animals.

The skull and crossbones symbol is universally recognized as
representing toxicity, which corresponds to **Division 6.1** of
Class 6 hazardous materials. <answer> \boxed{Class 6} </answer>
```

**Reference Answer:**

```
Class 6
```

---

**Example question in training dataset (category: Physics)**

**Prompt:**

```
<|im_start|>user
```

---

```
For what type of object is the period-luminosity relation used for
determining distances?
Please reason step by step, and put your final answer within
<answer> \boxed{} </answer>.<|im_end|>
<|im_start|>assistant
```

**Our Model's Response:**

The period-luminosity relation is a fundamental tool in astronomy
used to determine distances, particularly for a specific type of
star known as Cepheid variables.

Cepheid variables are pulsating stars that exhibit a regular
pattern of expansion and contraction. This pulsation causes them
to vary in luminosity, and the key insight about these stars is
that the period of
their luminosity variation is directly related to their intrinsic
luminosity (absolute magnitude). In other words, the longer the
period of pulsation, the more intrinsically luminous the star is.

This period-luminosity relation was first discovered by Henrietta
Leavitt in 1908 by studying Cepheid variables in the Large
Magellanic Cloud. Because Cepheids are so bright intrinsically,
they can be used as "standard candles" to measure distances within
our galaxy and to nearby galaxies.

Here's how the period-luminosity relation is used to determine
distances:

1. **Observe the Period**: First, astronomers measure the period
of the star's brightness variation, which is the time it takes for
the star to go from brightest to dimmest and back to brightest.

2. **Determine Intrinsic Luminosity**: Using the period-luminosity
relation, they can then determine the star's intrinsic luminosity.
This relation is usually given in a form where a graph or a
formula relates period to absolute magnitude (a measure of
intrinsic brightness).

3. **Compare to Apparent Luminosity**: The apparent magnitude (how
bright the star appears from Earth) can be measured directly.

4. **Calculate Distance**: Using the inverse square law for light,
which relates the intrinsic luminosity, apparent brightness, and
distance, astronomers can then calculate the distance to the
Cepheid variable.

This method is crucial for measuring distances in astronomy
because it provides a way to gauge how far away these stars are
with a high degree of accuracy. The distances determined using
Cepheid variables are themselves used to calibrate other distance
measurement techniques, such as those involving supernovae.

Given these details, the type of object for which the
period-luminosity relation is used for determining distances is
Cepheid variables. <answer> \boxed{Cepheid variables} </answer>

**Reference Answer:**

cepheid variable stars

---

**Example question in MMLU-Pro benchmark (category: Psychology)**

**Prompt (Here we ignore the prompt template and only show the question for brevity):**

```
During the second stage of Kohlberg's preconventional level of
moral development, children obey rules because:
Options are:
A. they are taught that rules are unbreakable.
B. they enjoy following the rules.
C. they are rewarded for following rules.
D. they feel they have a personal duty to uphold rules and laws.
E. doing so helps them satisfy their personal needs.
F. they understand the concept of fairness and justice.
G. doing so helps them avoid punishment.
H. everyone else is doing it.
I. they believe in the inherent goodness of rules.
J. they fear societal disapproval.
```

**Our Model's Response**:

```
In Kohlberg's theory of moral development, the preconventional
level is the first stage of moral reasoning, and it is primarily
focused on self-interest. This level is divided into two stages:

1. **Stage 1: Obedience and Punishment Orientation**
   In this stage, children obey rules to avoid punishment. Their
   reasoning is fear-based, and they view rules as set by
   authority figures. The focus is on the consequences of
   following or breaking rules, specifically the avoidance of
   punishment.

2. **Stage 2: Individualism and Exchange**
   In this stage, which is the second stage of the
   preconventional level, children recognize that different
   people have different needs and interests. They obey rules
   only when it is in their own interest or when they can benefit
   from doing so. This stage is often described as the "what's in
   it for me?" stage. Children at this level understand the
   concept of exchange and may follow rules if it leads to a
   reciprocal benefit.

Now, let's evaluate the options:

(A) they are taught that rules are unbreakable.
    - This does not align with either stage of the preconventional
    level.

(B) they enjoy following the rules.
    - Enjoyment of rules is more characteristic of conventional
    levels of moral development.

(C) they are rewarded for following rules.
    - While this could be somewhat related to Stage 2, the primary
    focus in Stage 2 is on personal benefit rather than external
    rewards.

(D) they feel they have a personal duty to uphold rules and laws.
    - This is more aligned with the conventional level,
    specifically the intermediate stage.

(E) doing so helps them satisfy their personal needs.
    - This could be seen as related to Stage 2, but it is not the
    most precise description.

(F) they understand the concept of fairness and justice.
```

```
        - Understanding fairness and justice is more associated with
        the conventional and post-conventional levels.

(G) they avoid punishment.
        - This describes Stage 1 of the preconventional level, not
        Stage 2.

(H) everyone else is doing it.
        - This is more related to conformity and is not a key
        characteristic of Kohlberg's stages.

(I) they believe in the inherent goodness of rules.
        - This is more characteristic of the post-conventional level.

(J) they fear societal disapproval.
        - Fear of societal disapproval is more related to the
        conventional level.

The best fit for Stage 2 of the preconventional level is (E) doing
so helps them satisfy their personal needs, as it reflects the
idea of mutual benefit and personal interest. However, (G) they
avoid punishment describes Stage 1 more accurately. Given the
typical interpretations of Kohlberg's stages, (E) is the more
appropriate choice for Stage 2.
=
But let's double-check: According to Kohlberg, in Stage 2,
children's moral reasoning is based on the notion of exchange and
the satisfaction of self-interest. Therefore, (E) is the most
accurate option.

<answer> \boxed{E} </answer>
```

**Base Model's Response:**

```
During the second stage of Kohlberg's preconventional level of
moral development, children obey rules because they understand the
concept of fairness and justice. This stage is characterized by a
focus on the consequences of actions and the desire to avoid
punishment or gain rewards. Children at this stage begin to
recognize that rules are not absolute and can be negotiated based
on the situation and the needs of the individual. They start to
consider the perspectives of others and the fairness of the rules,
which is a significant step towards more advanced moral reasoning.

Therefore, the correct answer is:

\boxed{F}
```

**Reference Answer**:

```
E
```

## H    LLM USAGE

We used an OpenAI LLM (GPT-5) as a writing and formatting assistant. In particular, it helped refine grammar and phrasing, improve clarity, and suggest edits to figure/table captions and layout (e.g., column alignment, caption length, placement). The LLM did not contribute to research ideation, experimental design, implementation, data analysis, or technical content beyond surface-level edits. All outputs were reviewed and edited by the authors, who take full responsibility for the final text and visuals.

