# OpenReview forum: "Reinforcing General Reasoning Without Verifiers"
_ICLR.cc/2026/Conference — ICLR 2026 Poster_

### Official Review · Reviewer_tixt · 2025-10-25

**Soundness:** 3
**Presentation:** 3
**Contribution:** 2
**Rating:** 4
**Confidence:** 4

**Summary:**

This paper introduces VeriFree, a method that extends R1-Zero-style reinforcement learning to general-domain, non-mathematical reasoning by using the model's own probability of the ground-truth answer as the reward signal, thereby eliminating the need for explicit verifiers. The authors evaluate VeriFree on broad reasoning benchmarks akin to GeneralReasoner and show that it outperforms a strong LM-based verifier baseline.

**Strengths:**

- The method is simple yet effective, offering a clear mathematical derivation and concise comparisons with existing algorithms.
- By tackling general-domain reasoning rather than confining itself to mathematics, this work addresses a pressing gap and distinguishes itself from the current wave of math-only RL studies.

**Weaknesses:**

- The VeriFree method uses implicit reward induced by model confidence, which is unavoidably an imperfect measure. Where budget is unconstrained, practitioners will still prefer a heavyweight but higher-fidelity verifier; the method therefore does not raise the performance ceiling of RL reasoning, it merely lowers the cost floor to some extent.
- All reported experiments rely on single-word or short-phrase ground-truth answers; leaving open the critical question of whether the method can be applied to problems with long answers and coding tasks.

**Questions:**

See weakness.

---

> ### Author Response · Authors · 2025-11-21
> **Rebuttal by Author**
>
> Thank you for your valuable review and suggestions. Below we respond to the comments in **Weaknesses (W)**.
>
> ---
>
> ***W1: Where budget is unconstrained, practitioners will still prefer a heavyweight but higher-fidelity verifier; the method therefore does not raise the performance ceiling of RL reasoning, it merely lowers the cost floor to some extent.***
>
> In most scenarios, VeriFree is preferred due to its slightly superior performance and significantly higher computational efficiency. However, in ideal cases where a highly accurate, robust, unbiased verifier without reward hacking issues is available and computational resources are unconstrained, one could potentially combine VeriFree's model confidence approach with such a verifier for enhanced results. This hybrid approach would utilize the probability of generating answers deemed correct by the verifier while maintaining the lower variance gradient estimation benefit of our method. We plan to explore this integration of verifier-based rewards with model probability as future work.
>
> ---
>
> ***W2: All reported experiments rely on single-word or short-phrase ground-truth answers; leaving open the critical question of whether the method can be applied to problems with long answers and coding tasks..***
>
> In our main experiments, the dataset preprocessing retains only short-phrase reference answers with token lengths under 7. To investigate whether VeriFree can effectively handle longer answers, we conduct an additional experiment using the training data from General Reasoner [1], which includes examples with longer reference answers. We align the training and evaluation protocols with General Reasoner, and denote this variant as "Qwen3-4B-Base-VeriFree w/ GR data". The results are presented below:
>
> |  | GPQA-Diamond | MMLU-Pro | SuperGPQA |
> |---|---|---|---|
> | Qwen3-4B-Base-Verifier | 44.4 | 63.0 | 34.3 |
> | Qwen3-4B-Base-VeriFree w/ our data | 42.4 | 63.5 | 35.1 |
> | General-Reasoner-4B | 42.9 | 62.8 | 32.5 |
> | Qwen3-4B-Base-VeriFree w/ GR data | 45.5 | 63.2 | 34.5 |
>
>
> Even when trained on data containing longer reference answers, VeriFree continues to outperform the baselines. It is also noteworthy that our method remains significantly more computationally efficient: the "Qwen3-4B-Base-VeriFree w/ GR data" variant requires approximately 32 GPU hours for training, whereas General-Reasoner-4B requires about 64 GPU hours (trained on 4 nodes with 8×H100 GPUs for 2 days).
>
> VeriFree is not well-suited for code generation tasks, as such datasets typically provide test cases rather than reference code solutions. This represents a limitation of our approach, which relies on the availability of question–answer pairs for training. We have already discussed this limitation in Appendix F.2.
>
>
> **References:**\
> [1] General-Reasoner: Advancing LLM Reasoning Across All Domains, NeurIPS 2025

---

### Official Review · Reviewer_2Bnh · 2025-10-26

**Soundness:** 2
**Presentation:** 3
**Contribution:** 3
**Rating:** 2
**Confidence:** 3

**Summary:**

This paper proposes VeriFree, a verifier-free methodology for general reasoning tasks beyond rule-based verification. The authors explicitly test VeriFree on a variety of domains such as economics, chemistry, and health (among others) to examine performance on multi-task understanding, and `graduate student level reasoning'. In addition, for the case when there is exactly one correct answer to questions, the authors provide theoretical analysis demonstrating that the approach of VeriFree has lower variance than methods which use a verifier, including reinforcement learning with verified rewards (RLVR).

**Strengths:**

(+) The paper presents an interesting approach to bypass the need for verification for more general reasoning tasks that might not be rule-based.

(+) The claims are supported by analysis (for one specific setting- also see Weaknesses, below) and experimental evaluations.

(+) The performance of VeriFree is comparable to verifier-based methods on a large variety of tasks, providing a promise of improved computational efficiency resulting from eliminating reliance on a strong verifier.

(+) Sec. 2.3 comparing the gradient estimator in VeriFree to estimators from alternate approaches offers particularly helpful and revealing insight into the working of VeriFree (for the single correct answer setting).

**Weaknesses:**

(-) The authors claim that VeriFree is the first verifier-free methodology for this class of problems. It is not clear if benchmarking against verifier-based methods only adequate.

(-) What does `rule-based answer verification’ exactly mean? It is reasonable to conjecture that some of the domains mentioned might support rule-based answer verification (e.g., based on statutes of laws). This question becomes even more relevant since the authors state in the Limitations (in the Appendix) that VeriFree is constrained by its dependency on question-answer pairs for supervision.

(-) It is not clear if the single answer assumption is a reasonable one to make for domains beyond rule-based verification. Specifically, one would assume the potential ambiguities in answers to questions in these domains is a characteristic that has necessitated use of a verifier.

(-) From Fig. 1, the claim of VeriFree surpassing instruct models and models tuned with a specialized LLM verifier are somewhat strong, given that the accuracy numbers in the bar graphs for VeriFree are only very marginally higher in 5 out of 6 graphs and comes in second best in one graph. At best, one can conclude that VeriFree matches the accuracy performance of these models.

(-) In Line 107, the paper claims that VeriFree is simpler, faster, less memory-intensive, and more robust than verifier-based alternatives. It is not clear which experiments in the paper correspond to evaluations of memory intensiveness and robustness.

(-) In Tables 1 and 2, it is not clear to me how the quantity in the Avg column is computed. Checking the numbers, it does not appear to be equal to the mean (average) of the numbers in the same row in the columns on the right. For example, in Table 1, for Qwen3-8B Base-VeriFree, the Avg reported is **67.2**, while **mean(71.5,85.3,73.5,…,45.6) = 66.3**. If it is indeed the case that the Avg should correspond to the mean of the entries in the same row, then this discrepancy extends to many of the other numbers in both tables.

(-) While VeriFree performs better when looking at the average accuracy, for the 4B experiments in Table 1, VeriFree obtains a higher accuracy than Verifier in only 8 out of 14 domains. Similarly, for the 8B experiments in Table 2, VeriFree obtains higher accuracy that Verifier in only 7 out of 13 domains.

(-) Minor typo: Line 183 - marginalizes —> marginalize.

**Questions:**

The authors' comments on the points raised in the Weaknesses section, above, will be helpful.

---

> ### Author Response · Authors · 2025-11-21
> **Rebuttal by Author (1/2)**
>
> Thank you for your valuable review and suggestions. Below we respond to the comments in **Weaknesses (W)**.
>
> ---
>
> ***W1: It is not clear if benchmarking against verifier-based methods only adequate.***
>
> To the best of our knowledge, our work presents the first verifier-free method for this task. While label-free approaches exist in related areas, they address different problem settings and are not directly comparable. The most relevant and important baseline for our method is verifier-based approaches for general-domain reasoning. Our benchmarking results demonstrate that our method not only slightly outperforms these verifier-based approaches but also does so with computational efficiency.
>
> ---
>
> ***W2: What does "rule-based answer verification" exactly mean?***
>
> Rule-based answer verification refers to systems that can (almost) perfectly determine the correctness of an LLM-generated answer by relying on deterministic rules. For instance, in mathematics, systems like Math-Verify can evaluate the equivalence of expressions, and in code generation, outputs can be compiled and tested against predefined test cases. These systems are highly robust and reliable.
>
> However, to the best of our knowledge, such robust rule-based verifiers are unavailable for the general domains that are the primary focus of our work. We welcome specific counterexamples if they exist. In their absence, the widely accepted alternative is to use model-based verifiers, which fall outside the category of "rule-based" systems and introduce a different set of challenges, as discussed in our paper.
>
> ---
>
> ***W3: It is not clear if the single-answer assumption is a reasonable one to make for domains beyond rule-based verification.***
>
> This issue is related to the base model's strong generalization capability, which includes a "reasoning prior" acquired during pre-training. This prior encompasses an understanding of logical structures and problem-solving patterns. The single reference answer does not supply reasoning ability but acts as a precise alignment signal that activates these latent capabilities. It demonstrates the target reasoning methodology, enabling the model to extract and generalize the underlying principle rather than memorizing superficial patterns. This makes the learning process highly sample-efficient.
>
> This claim is also supported by our ablation study on mathematical data, where we compare versions of VeriFree using single versus multiple reference answers. The results (Figure 6, right) show that while multiple references yield modest gains, the performance difference is limited. This suggests that a single high-quality reference provides a sufficiently strong signal, with rapidly diminishing returns from additional references.

---

> ### Author Response · Authors · 2025-11-21
> **Rebuttal by Author (2/2)**
>
> ***W4: The claim of performance superiority is too strong.***
>
> We acknowledge that our method achieves a slight performance gain over the baseline that relies on a model-based verifier. This result still highlights a key advantage of our approach: it attains slightly superior performance without requiring an additional tuned verifier model, leading to a simpler and more computationally efficient training process.
>
> ---
>
> ***W5: It is not clear which experiments in the paper correspond to evaluations of memory intensiveness and robustness.***
>
> The memory overhead of the verifier-based method is obvious, as it requires loading an additional 1.5B parameter model to serve as the inference engine for verification during training the reasoning model.
>
> While robustness is difficult to quantify directly, VeriFree's superior performance, achieved using only a single reference answer per question, demonstrates considerable robustness compared to the verifier-based baseline. Furthermore, model-based verifiers are susceptible to issues like reward hacking and biases (e.g., length bias). For instance, the baseline method relies on an answer token length penalty (Lines 353-356) to prevent degeneration, a heuristic our method does not require. By avoiding the need for such corrective measures, VeriFree is inherently more robust.
>
> ---
>
> ***W6: How the quantity in the Avg column is computed.***
>
> The reported average is a micro-average. This metric aggregates the contributions across all classes to compute a global measure, rather than computing the metric per class and then averaging (which would assign equal weight to each class).
>
> ---
>
> ***W7: VeriFree does not outperform the verifier-based method in all domains.***
>
> We acknowledge that VeriFree does not outperform all verifier-based methods in every domain. However, it is important to note that our method achieves higher overall accuracy than the baseline approach. Moreover, our method is more efficient as it eliminates the need to train a separate model-based verifier and avoids the computational cost of sampling from such a verifier during the training of the reasoning model.
>
> ---
>
> ***W8: Minor typos.***
>
> Thank you for highlighting these issues. We will correct the typos in the revised version.

---

> > ### Comment · Reviewer_2Bnh · 2025-11-21
> > **Thank you Authors**
> >
> > I thank the authors for their response. While I am satisfied with responses to W1 - W3, I am not fully convinced by responses to W4 - W7.
> >
> > The response to W4 suggests that the original claim of VeriFree significantly surpassing performance in Fig. 1 has now been updated to claiming that VeriFree matches the performance with an additional benefit of not requiring an additional tuned verifier model.
> >
> > The response to W5 indicates that the robustness is a function of answer length. However, one could also claim that VeriFree is less robust in the context of relying on situations when there is exactly one correct answer. I am also not fully convinced by the authors' response that robustness is difficult to quantify directly- quantifiable indicators of robustness have been widely used across different domains to demonstrate improved performance.
> >
> > The response to W6 still does not provide detail on how the reported average is computed. In my review, I had provided a specific example from one of their experiments where I believed the numbers did not match. A reader will benefit from details on exactly how this 'micro average' is computed.
> >
> > The response to W7 suggests that improved average performance across domains is preferable to improved performance in specific domains. I can see the authors' perspective on this point, and will give them the benefit of the doubt, despite being not fully satisfied that VeriFree is not the best performing in more than 50% of the domains.
> >
> > Overall, I think that many of the claims of superior performance of VeriFree should highlight the fact that additional training of model-based verifiers is not needed, and that VeriFree **matches** the performance of verifier-based methods in multiple domains.

---

> ### Author Response · Authors · 2025-11-22
> **Follow-up Response by Author (1/2)**
>
> Thanks for your further comment. Below we respond to the further comments in **Weaknesses (W)**.
>
> ---
>
> ***W4: The original claim has been updated to the claim that VeriFree matches the performance with an additional benefit of not requiring an additional tuned verifier model.***
>
> We think the claim should consider the choice of evaluation on a case-by-case basis. The primary large-scale benchmarks for general-domain reasoning are MMLU-Pro (12,032 samples) and SuperGPQA (26,529 samples), and we focus on overall average accuracy across these benchmarks.
>
> For the 1.7B model, VeriFree matches the performance of the verifier-based method on both benchmarks. For the 4B model, VeriFree shows comparable performance on MMLU-Pro and superior performance on SuperGPQA. For the 8B model, VeriFree demonstrates advantages on both benchmarks.
>
> We attribute this scaling pattern to our method's use of the model's own confidence in the reference answer as the training signal. As the base model size increases, this confidence estimate becomes more accurate. In contrast, the verifier-based method uses a fixed 1.5B model-based verifier, whose accuracy does not scale with the reasoning model's capacity.
>
>
> ---
>
> ***W5: The response to W5 indicates that the robustness is a function of answer length.***
>
> To provide further clarification on robustness concerns, we offer a more detailed explanation below.
>
> We cited length bias as one example of the inherent limitations in model-based verifiers. This form of bias (where verifiers tend to favor longer answers) has been widely observed in reward modeling. In extreme cases, lengthy but incorrect answers may be viewed as correct by the model-based verifier. Following General Reasoner [1], we incorporate a length penalty to mitigate this issue by penalizing answers whose token length significantly diverges from the reference answer. Without this correction, verifier-based methods exhibit noticeable performance degradation.
>
> Length bias represents just one manifestation of broader robustness issues. There are many other observed biases in LLM-as-a-judge or model-based verifiers [2,3]. Another example is lexical bias, where verifiers favor specific keywords regardless of actual correctness. Such phenomena illustrate fundamental limitations in model-based verifiers. In contrast, VeriFree inherently avoids these biases by using the model's own probability of generating the reference answer, a metric unaffected by extraneous factors like length or specific vocabulary, thus ensuring greater robustness.
>
> Theoretically, a perfect verifier that recognizes all valid answers should outperform methods using a single reference answer per question. To test this, we compared VeriFree trained on math data with single versus multiple reference answers (Figure 6, Right). Results confirm that multiple reference answers yield modest gains, supporting the theoretical expectation.
>
> Notably, however, VeriFree currently matches or exceeds verifier-based performance despite using only a single reference answer per question. We attribute this to its more robust reward signal, which avoids verifier-induced biases. This advantage is particularly evident in larger models, where the base model's confidence becomes increasingly reliable.
>
> **References:**\
> [1] General-Reasoner: Advancing LLM Reasoning Across All Domains, NeurIPS 2025\
> [2] Justice or prejudice? quantifying biases in llm-as-a-judge. ICLR 2025\
> [3] Humans or llms as the judge? a study on judgement biases. EMNLP 2024

---

> ### Author Response · Authors · 2025-11-22
> **Follow-up Response by Author (2/2)**
>
> ***W6: The response to W6 still does not provide detail on how the reported average is computed.***
>
> The reported average accuracy is computed by taking the mean over all test samples (i.e., a micro-average), rather than first averaging within each category and then averaging across categories. This approach better reflects overall performance given the imbalanced sample sizes across different domains. Similar evaluation protocols are used in other works such as General Reasoner [1] (see their open-sourced evaluation code for reference: https://github.com/TIGER-AI-Lab/General-Reasoner/blob/main/evaluation/eval_supergpqa.py#L111).
>
> The total number of test samples in MMLU-Pro and SuperGPQA is 12,032 and 26,529, respectively. The number of samples across domains in these benchmarks is summarized in the table below (both benchmarks are publicly available, and detailed statistics can be readily verified).
>
> | MMLU-Pro domains | Num samples | SuperGPQA domains | Num samples |
> |---|---|---|---|
> | computer science | 410 | Engineering | 7892 |
> | math | 1351 | Medicine | 2755 |
> | chemistry | 1132 | Science | 9838 |
> | engineering | 969 | Philosophy | 347 |
> | law | 1101 | Military Science | 205 |
> | biology | 717 | Economics | 873 |
> | health | 818 | Management | 501 |
> | physics | 1299 | Sociology | 143 |
> | business | 789 | Literature and Arts | 1676 |
> | philosophy | 499 | History | 674 |
> | economics | 844 | Agronomy | 485 |
> | other | 924 | Law | 656 |
> | psychology | 798 | Education | 484 |
> | history | 381 |  |  |
>
>
> We take experiments with the 4B model scale as an example:
>
> For Qwen3-4B-Base-Verifier:
>
> MMLU-Pro average accuracy:\
> (410 * 66.1 + 1351 * 81.3 + 1132 * 69.7 + 969 * 52.8 + 1101 * 29.1 + 717 * 79.8 + 818 * 62.8 + 1299 * 67.6 + 789 * 71.2 + 499 * 48.5 + 844 * 73.1 + 924 * 52.8 + 798 * 68.5 + 381 * 45.4) / 12032 = 63.02
>
> SuperGPQA average accuracy:\
> (7892*35.4 + 2755*35.5 + 9838 * 34.5 + 347 * 39.2 + 205 * 41.0 + 873 * 39.1 + 501 * 36.7 + 143 * 37.1 + 1676 * 26.6 + 674 * 26.6 + 485 * 33.8 + 656 * 33.1 + 484 * 35.3) / 26529 = 34.348
>
>
> For Qwen3-4B-Base-VeriFree:
>
> MMLU-Pro average accuracy:\
> (410 * 64.4 + 1351 * 82.2 + 1132  * 70.1 + 969 * 55.6 + 1101 * 30.7 + 717 * 81.7 + 818 * 59.2 + 1299 * 71.0 + 789 * 71.0 + 499 * 47.1 + 844 * 71.7 + 924 * 53.4 + 798 * 66.8 + 381 * 47.5) / 12032 =63.539
>
> SuperGPQA average accuracy:\
> (7892 * 36.3 + 2755 * 34.5 + 9838 * 36.9 + 347 * 35.7 + 205 * 37.1 + 873  * 39.1 + 501 * 38.3 + 143 * 31.5 + 1676 * 24.7 + 674 * 22.0 + 485 * 33.0 + 656 * 33.2 + 484 *  34.1) / 26529 = 35.165
>
> We hope this clarification aids in understanding our evaluation protocols.
>
>
>
>
>
> ---
>
> ***W7: Claims about performance across all domains and specific domains.***
>
> Average accuracy is the main metric, since some specific domains of special interest might have very limited test samples. For example, there are only 205 test samples that belongs to "Military Science" domain in SuperGPQA benchmark.
>
> So, overall, the most accurate clain could be VeriFree matches or exceeds the methods with model-based verifiers in terms of average performance, and it cannot exceed the baseline in every domain. And notable, the model-based verifier needs additional data and computational cost to train, and RL with the model-based verifier is a very strong baseline that requires more engineering effort and computational cost.
>
> Average accuracy serves as the primary evaluation metric, as certain specialized domains may contain very few test samples. For instance, the "Military Science" domain in the SuperGPQA benchmark comprises only 205 samples.
>
> Overall, the most accurate conclusion is that VeriFree matches or exceeds the performance of model-based verifier methods in terms of average accuracy, though it may not surpass the baseline in every individual domain. It is important to note that the model-based verifier baseline requires additional training data and substantial computational resources. Given that RL with a model-based verifier is a strong and computationally intensive baseline, the competitive performance of VeriFree underscores its efficiency and practicality.
>
>
> ---
>
> Thank you for recognizing the contribution of our work and for your valuable feedback. If you have any further concerns, please let us know. If our responses have addressed your questions, we would be grateful if you could reconsider your assessment of our submission.

---

> > ### Comment · Reviewer_2Bnh · 2025-11-22
> > **Thank you**
> >
> > Thank you, authors for your detailed response. I have increased my score by 2 points. I have no further comment.

---

> > > ### Author Response · Authors · 2025-11-23
> > > **Thank you for your feedback**
> > >
> > > Thank you for your timely response and for updating the score. As a score of 4 is still considered negative, could you please clarify any remaining concerns or issues that we may not have addressed adequately? Your feedback would be very valuable for us to further improve the work.

---

> > > > ### Comment · Reviewer_2Bnh · 2025-11-23
> > > > **Reply to Authors**
> > > >
> > > > Thank you authors. I respectfully disagree. A score of 4 indicates that I would not mind if the paper is accepted. Between the initial submission that was reviewed and the response, there was significant additional experiments and clarifications.
> > > >
> > > > In this light, I believe that the overall paper in its current form might benefit from a full review, with all the additional results included. However, if another reviewer (or reviewers) is willing to champion the paper for acceptance, I will not object. I wish you the best.

---

### Official Review · Reviewer_r1nM · 2025-11-02

**Soundness:** 3
**Presentation:** 3
**Contribution:** 3
**Rating:** 4
**Confidence:** 3

**Summary:**

This paper propose a novel VeriFree framework that bypass answer verification and assign rewards according to the probability of generating reference answer.

Unique answer assumption might not hold generally.

**Strengths:**

1. The proposed VeriFree framework is simple to implement and bypass the need of verifiers.
2. Experimental results demonstrate that VeriFree achieves comparable results to the verifier-based baseline and shows good transferable reasoning skill gains.

**Weaknesses:**

1. Evaluation reliability: Single run pass@1 results are too noisy for small benchmarks like MATH-500, Minverva etc. The author might have to report Avg@k and other statistics for robust demonstration.
2. Missing baselines: many existing probability- or frequency-based baselines are missing, such as TTRL.
3. The assumption that only single accurate answer exists might be problematic due to the flexibility of language (e.g., 'A is larger than B' and 'B is the smaller one'). The author might need to conduct more analysis on this assumption.

**Questions:**

1. Does VeriFree performs better than existing methods that rely on verifiers such as DAPO and ORZ. As RLVR exhibits good generalization capability, could it be possible that training on mathematical problems can yields better results?
2. How does VeriFree works on other model families, considering the data contamination risk of Qwen3 model family.

---

> ### Author Response · Authors · 2025-11-21
> **Rebuttal by Author**
>
> Thank you for your valuable review and suggestions. Below we respond to the comments in **Weaknesses (W)** and **Questions (Q)**.
>
> ---
>
> ***W1: Evaluation reliability.***
>
> Thank you for pointing this out. We note that the primary benchmarks in our evaluation are large in scale. For instance, MMLU-Pro includes 12k test samples and SuperGPQA contains 26k. On these large benchmarks, VeriFree generally achieves stronger performance.
>
> We have also added evaluations on smaller benchmarks, as shown in the table below. We report Avg@K, where K is chosen such that the total number of samples after repetition is approximately 1k.
>
> |  | GPQA-Diamond (Avg@4) | AIME24 (Avg@32) | MATH500 (Avg@2) | Minerva (Avg@4) |
> |---|---|---|---|---|
> | Qwen3-4B-Base-Verifier | 42.80 | 16.04 | 73.70 | 24.63 |
> | Qwen3-4B-Base-VeriFree | 42.29 | 17.50 | 75.10 | 25.54 |
> | Qwen3-8B-Base-Verifier | 44.07 | 18.96 | 77.20 | 36.50 |
> | Qwen3-8B-Base-VeriFree | 45.45 | 25.52 | 81.10 | 34.67 |
>
> ---
>
> ***W2: Missing baselines: TTRL.***
>
> TTRL is not a baseline of VeriFree. TTRL [1] employs a frequency-based majority voting mechanism to determine the "reference answer", thereby eliminating the need for a pre-specified reference answer in the training data. In contrast, our method uses the conditional probability of generating the reference answer given the question and reasoning traces as a learning signal, which avoids dependence on a verifier but still requires a reference answer.
>
> **The two approaches are orthogonal and potentially complementary.** A promising direction for future work would be to combine them, which could further enhance performance in specific application scenarios.
>
> ---
>
> ***W3: The assumption that only single accurate answer exists might be problematic due to the flexibility of language.***
>
> It is related to the base model's strong generalization ability. Even when trained to produce answers in a specific style, it learns the underlying reasoning principles, similar to how models in supervised fine-tuning learn general instruction-following from stylized examples.
>
> We agree that an ideal verifier, which rewards semantic correctness regardless of style or expression, is a powerful concept. Our method, by maximizing the likelihood of the reference answer $\\pi_{\\theta}(y^*|x)$, is indeed more constrained. However, such ideal verifiers are often unavailable for general domains, and model-based alternatives can be prone to reward hacking and add significant overhead.
>
> Empirically, our results (Tables 1\&2) show that this constraint does not hinder performance. The base model successfully generalizes from the specific language style to the core task of producing correct reasoning, demonstrating effective knowledge transfer.
>
> ---
>
> ***Q1: Could it be possible that training on mathematical problems can yield better results?***
>
> Yes, we have compared VeriFree to rule-based verification methods on math training data. The results, presented in Table 4, show that the two methods achieve similar accuracy, with VeriFree performing marginally better.
>
> ---
>
> ***Q2: How does VeriFree work on other model families, considering the data contamination risk of Qwen3 model family?***
>
> We have extended our evaluation to an additional model family, applying both our method and the verifier-based baseline to OctoThinker-3B-Long-Base [2], a model mid-trained from Llama-3. The performance is compared against the official RLVR-trained reasoning model (OctoThinker-3B-Long-Zero) and the corresponding verifier-based baseline. All evaluations are conducted under a zero-shot setting. Results are presented below:
>
> |  | MMLU-Pro | SuperGPQA | GQPA-D |
> |---|---|---|---|
> | OctoThinker-3B-Long-Base | 5.2 | 2.1 | 0 |
> | OctoThinker-3B-Long-Zero | 35.8 | 18.6 | 25.3 |
> | OctoThinker-3B-Long-Verifier | 35.5 | 19.8 | 28.3 |
> | OctoThinker-3B-Long-VeriFree | 36.2 | 20.5 | 30.3 |
>
> Our model can outperform both the official RLVR model and our implemented counterpart with a model-based verifier. The base model does not perform well due to poor instruction-following abilities.
>
> **References:**\
> [1] TTRL: Test-Time Reinforcement Learning, 2025\
> [2] Octothinker: Mid-training incentivizes reinforcement learning scaling, 2025

---

### Official Review · Reviewer_9QRi · 2025-11-06

**Soundness:** 2
**Presentation:** 3
**Contribution:** 3
**Rating:** 6
**Confidence:** 4

**Summary:**

The paper proposes an RL framework called VeriFree, which extends the R1-Zero paradigm (DeepSeek-R1-Zero) to general reasoning tasks without requiring explicit verifiers. Existing methods like RL with verifiable rewards (RLVR) rely on rule-based or model-based verifiers (often another LLM), which are infeasible for open-ended domains such as law, medicine, or economics. VeriFree addresses this by eliminating the verifier entirely while preserving the core benefits of RL-based fine-tuning. In R1-Zero-style training, a model generates a reasoning trace (z) and a final answer (y). A verifier gives a reward = 1 if y matches the reference answer y*, else 0. VeriFree reformulates this process: instead of relying on a verifier, it directly maximizes the model’s probability of generating the reference answer. The resulting gradient estimator combines: (a) a reasoning term weighted by the model’s likelihood of producing the correct answer, and (b) a supervised term weighted by the same likelihood.

This makes it equivalent in expectation to RLVR under single-correct-answer cases but with lower gradient variance (proved via Rao-Blackwellization). The approach integrates variance reduction (RLOO) and tokenization-aware patching to ensure stable optimization.

For experimental evaluation, Qwen3 (1.7B, 4B, 8B) base models are fine-tuned with VeriFree using the Oat framework. Benchmarks used are MMLU-Pro, GPQA, SuperGPQA (for general reasoning) and MATH-500, OlympiadBench, Minerva Math, GSM8K, AMC, AIME24 (for mathematical reasoning). On MMLU-Pro and SuperGPQA, VeriFree matches or surpasses verifier-based RL baselines while being simpler and more compute-efficient. The accuracy gains are +12-40% over base models, and it achieves faster convergence and higher final accuracy than verifier-based RL.

**Strengths:**

- The paper is very well-written and structured well.

- The concept of verifier-free RL eliminates reliance on rule-based or LLM verifiers, making RL training scalable to open-ended reasoning domains. Further, it is derived directly from the RL objective; proven equivalence to RLVR under certain assumptions with formal variance reduction.

- Strong empirical results: Matches or surpasses verifier-based methods across multiple benchmarks while being simpler, faster, and less memory-intensive. The proposed method demonstrates reasoning transfer to unseen domains and improved convergence behavior.

**Weaknesses:**

- In Tables 1 and 2, it is unclear why the accuracy of Qwen3-4B-Base-Verifier is lower than that of Qwen3-4B-Base-VeriFree. In VeriFree, the LLM output is first parsed into reasoning tokens and a generated final answer (y). Then, the generated answer is replaced with the gold-standard final answer (y⁎). If only one answer (y⁎) is correct and receives a reward of 1 (while all others receive 0), then the expected reward for a reasoning trace z can be computed directly as the probability assigned to y⁎, effectively marginalizing out y. In contrast, when using a verifier, y⁎ will receive a reward of 1 from the verifier, but all other correct answers (for questions with multiple valid answers) will also receive a reward of 1. Conceptually, this should make the verifier-based approach perform better than VeriFree, as it trains the model to obtain a reward of 1 for all correct answers rather than a single reference answer.

- The paper mentions that "when multiple valid answers exist, we show empirically that using just one as a reference provides a sufficient learning signal to elicit strong reasoning behavior."" However, it is not clear why this should be the case. In the VeriFree framework, the model is trained to maximize the reward only for a single correct answer.

- The theoretical equivalence holds only when there is a unique correct answer, limiting its applicability to open-ended or multi-valid-response tasks. In such cases, the model may overfit to the reference answers instead of exploring diverse reasoning paths, thereby reducing interpretability.

- For the baseline, it is unclear why a model-based (i.e., LLM-judge) reward model was used. For math and code tasks, rule-based or test-case-based evaluation is feasible and far more reliable than an LLM judge. The paper claims that VeriFree demonstrates strong transfer to math benchmarks; therefore, for math- or code-specific benchmarks, a stronger rule-based verifier should have been used as a baseline.

**Questions:**

See weaknesses.

---

> ### Author Response · Authors · 2025-11-21
> **Rebuttal by Authors**
>
> Thank you for your supportive review and suggestions. Below we respond to the comments in **Weaknesses (W)**.
>
> ---
>
> ***W1: It is unclear why the accuracy of Qwen3-4B-Base-Verifier is lower than that of Qwen3-4B-Base-VeriFree. In VeriFree.***
>
> We attribute this outcome to three main reasons: (1) Using a single reference answer instead of multiple answers does not significantly degrade performance. To quantify this effect, we conducted experiments on math training data using two versions of VeriFree: one with a single reference answer and another with multiple answers. The results (Figure 6, right) indicate that while multiple answers provide modest improvements, the performance gain is limited. (2) VeriFree offers provably lower variance in gradient estimation (Section 2.2 and Appendix B.2), which may lead to convergence to a better optimum. (3) The Qwen3-4B-Base-Verifier relies on a model-based verifier from [1], which is susceptible to reward hacking (e.g., length bias) [2,3] and may exhibit lower accuracy compared to rule-based verifiers (though the latter are only feasible in constrained domains like math). These factors may collectively hinder verifier-based methods.
>
> Considering these points, VeriFree outperforms verifier-based approaches even when using only a single reference answer, whereas verifier-based methods theoretically benefit from accommodating multiple correct answers.
>
> ---
>
> ***W2: Why does just one reference answer provide a sufficient learning signal to elicit strong reasoning behavior?***
>
> Our conjecture centers on the base model's strong generalization capability. The model possesses a ``reasoning prior'' from pre-training: it already understands logical structures and problem-solving patterns. The single reference answer serves not as the source of reasoning ability, but as a precise alignment signal that activates these latent capabilities. It demonstrates the desired methodology, allowing the model to extract and generalize the underlying principle rather than memorizing the specific answer. This makes the learning signal highly efficient and effective.
>
> ---
>
> ***W3: Trained on unique correct answer, the model may overfit to the reference answers instead of exploring diverse reasoning paths.***
>
> This concern can be understood through the lens of supervised fine-tuning (SFT). In standard SFT practice, each training question typically has only one reference answer, yet the trained model still produces diverse outputs. The key insight is that generalization depends more on the diversity of the input questions than on the multiplicity of reference answers per question. When the training corpus contains sufficiently diverse questions, overfitting is naturally mitigated. This explains why VeriFree maintains diverse reasoning paths despite each question having a single reference answer, and the variety in the training questions themselves promotes reasoning diversity.
>
> ---
>
> ***W4: It is unclear why a model-based (i.e., LLM-judge) reward model was used.***
>
> As outlined in Section 1, our work focuses on general-domain reasoning beyond mathematical and coding tasks. While methods like RLVR have demonstrated strong performance in domains with robust rule-based verifiers (e.g., mathematics and code), reasoning in general domains (where such verifiers are unavailable) remains under-explored. Following General Reasoner [1], which also targets general-domain reasoning and employs a model-based verifier, we adopt a similar approach as our main baseline.
>
> **References:**\
> [1] General-Reasoner: Advancing LLM Reasoning Across All Domains, NeurIPS 2025\
> [2] Justice or prejudice? quantifying biases in llm-as-a-judge. ICLR 2025\
> [3] Humans or llms as the judge? a study on judgement biases. EMNLP 2024

---

### Author Response · Authors · 2025-12-01
**General Response**

We thank the area chair for handling our submission and the reviewers for their time and valuable comments.

We appreciate that **all reviewers recognize our main contribution: VeriFree matches or outperforms verifier-based methods across multiple benchmarks  while being simpler, faster, and more memory-efficient.**

Below, we address the key points that reviewers raised (corresponding $\\textrm{\\color{blue}revisions}$ are highlighted in $\\textrm{\\color{blue}blue}$ in the manuscript):

***1. Concerns about the single reference answer assumption.*** (Reviewer 9QRi, r1nM, and 2Bnh)

As shown in **Tables 1&2**, VeriFree with a single reference answer matches or exceeds verifier-based methods that accommodate multiple correct answers. **Figure 6 (Right)** confirms that multiple references provide limited additional benefit.

This effectiveness stems from the model's strong generalization capability acquired during pretraining. The reference answer simply activates reasoning abilities already present in the model.

Training with a single reference answer per question does not cause diversity issues, analogous to standard SFT where each training question typically has only one reference answer, yet the trained model still produces diverse outputs. Our method extends SFT by incorporating reasoning traces.

***2. Evaluation reliability.*** (Reviewer r1nM)

We conduct new evaluations and report Avg@K for small benchmarks ($\\textrm{\\color{blue}Table 7}$).

***3. Experiments on other model families.*** (Reviewer r1nM)

We add new experiments on  OctoThinker-3B-Long-Base, a model mid-trained from Llama-3 ($\\textrm{\\color{blue}Table 8}$).

***4. Definition of "average" accuracy in Table 1&2.*** (Reviewer 2Bnh)

We use a micro average. The definition of micro average has been added to the revision ($\\textrm{\\color{blue}Line 347-351}$).

***5. Whether the method can be applied to problems with long answers.*** (Reviewer tixt)

We added experiments using the General Reasoner dataset (which has longer answers and no length filtering). VeriFree maintains strong performance on this dataset ($\\textrm{\\color{blue}Table 9}$).

---

### Meta-Review · Area_Chair_mME1 · 2026-01-06

**Summary:**

This paper introduces VeriFree, a verifier-free reinforcement learning approach that improves LLM reasoning by assigning rewards based on the log-likelihood of the reference answer given the generated reasoning, thereby eliminating the need for rule-based or LLM-based verifiers. The method optimizes an objective that is theoretically equivalent to verifier-based RL in expectation, while yielding lower variance and more stable training. Across general reasoning benchmarks, VeriFree matches or outperforms verifier counterparts that rely on offline LLM judges, while incurring lower cost and complexity. Overall, the paper challenges the assumption that external verifiers are necessary for improving reinforcement learning–based reasoning.


All reviewers agree that the paper proposes a simple yet interesting method, supported by well-motivated reasoning, including theoretical analysis and experimental results. The primary concerns raised by the reviewers and the AC largely align with the authors' summary and can be summarized as follows:

- Single-reference answer assumption and its performance implications: a key concern shared by multiple reviewers is that VeriFree fundamentally assumes the existence of a single correct reference answer, which may limit performance in settings where multiple valid answers exist.
- Experimental details and evaluation: reviewers raised several concerns regarding experimental design and reporting, including baseline choices and missing comparisons, evaluation reliability and statistical robustness, and the lack of evidence for generalization to long or complex-form outputs (e.g., coding tasks) as well as to other model families.

The authors can address the flagged hallucinated references in an autogenerated comment, which appears to stem from a typographical error in citing the "Qwen2.5-Math Technical Report".

**Reviewer Concerns:**

The AC finds that most reviewer concerns, including the above concerns, were adequately addressed through the rebuttal and additional experiments from the AC's perspective. Nevertheless, the AC notes that reviewers may still have concerns about the reported performance advantages over verifier-based methods. The paper's core contribution is indeed to demonstrate matching (or sometimes better) performance without relying on verifiers, rather than to achieve superiority over the methods using verifiers. The authors are encouraged to ensure a stronger alignment between the contributions and the empirical evidence. Furthermore, while the observed gains are small, the provably lower variance in gradient estimation alone may be insufficient to justify claims of superiority; so, additional supporting evidence would help strengthen the paper.  Furthermore, while the observed gains are small, the provably lower variance in gradient estimation alone may not be sufficient to justify the superiority - therefore, additional supporting evidence would help strengthen the paper. Taking these points into account,  the AC considers this paper an interesting work that provides useful insight for RLVR, and the final judgment is that the paper is sufficiently solid to meet the acceptance bar.

**Reviewer Scores:**

Reviewer 2Bnh increased their score from 2 to 4. Finally, all reviewers reached a borderline position, and from this AC's perspective, the rebuttal has addressed the raised concerns well. The AC therefore expects that at least one reviewer may further increase their rating.

---

### Decision · Program_Chairs · 2026-01-26

Accept (Poster)